# Reciprocal discoidin domain receptor signaling strengthens integrin adhesion to connect adjacent tissues

Kieop Park[1†], Ranjay Jayadev[1†], Sara G Payne[1,2†], Isabel W Kenny-Ganzert[1], Qiuyi Chi[1], Daniel S Costa[1], William Ramos-Lewis[1], Siddharthan B Thendral[1], David R Sherwood[1]*

[1]Department of Biology, Duke University, Durham, United States; [2]Department of Cell Biology, Duke University Medical Center, Durham, United States

*For correspondence:
david.sherwood@duke.edu

[†]These authors contributed equally to this work

Competing interest: The authors declare that no competing interests exist.

**Abstract** Separate tissues connect through adjoining basement membranes to carry out molecular barrier, exchange, and organ support functions. Cell adhesion at these connections must be robust and balanced to withstand independent tissue movement. Yet, how cells achieve synchronized adhesion to connect tissues is unknown. Here, we have investigated this question using the *Caenorhabditis elegans* utse-seam tissue connection that supports the uterus during egg-laying. Through genetics, quantitative fluorescence, and cell-specific molecular disruption, we show that type IV collagen, which fastens the linkage, also activates the collagen receptor discoidin domain receptor-2 (DDR-2) in both the utse and seam. RNAi depletion, genome editing, and photobleaching experiments revealed that DDR-2 signals through LET-60/Ras to coordinately strengthen an integrin adhesion in the utse and seam that stabilizes their connection. These results uncover a synchronizing mechanism for robust adhesion during tissue connection, where collagen both affixes the linkage and signals to both tissues to bolster their adhesion.

## eLife assessment

This **important** paper reveals how cells in adjacent tissues use the extracellular matrix to establish mechanical connections. Through a series of crisp genetic manipulations and quantitative image analyses, the authors provide **compelling** evidence to show how an essential adhesion between the uterus and the seam cells in the nematode *C. elegans* is formed. The assembly of type IV collagen triggers internalization of a cell surface receptor, which then signals from endocytic vesicles to strengthen the connection.

## Introduction

Basement membrane (BM) is a planar, cell-associated extracellular matrix (ECM) that underlies or surrounds most tissues. BM provides tissues with structural support, barrier functions, and signaling platforms (*Morrissey and Sherwood, 2015*; *Jayadev and Sherwood, 2017*; *Pozzi et al., 2017*). The two core BM matrix components are the heterotrimeric proteins laminin and type IV collagen. Laminin is comprised of an α, β, and γ subunit. Once secreted, laminin associates with cell surface receptors and self-assembles into a sheet-like network. Type IV collagen is a large protein comprised of two α1-like and one α2-like chains, and these heterotrimers cross-link to form a grid, which is thought to be connected to the laminin network through bridging molecules (*Hohenester and Yurchenco, 2013*). Type IV collagen provides BM with tensile strength, which allows BM to structurally support tissues and resist mechanical forces (*Fidler et al., 2018*). Type IV collagen is a ligand for several vertebrate

integrin receptors and the collagen-specific receptor tyrosine kinase (RTK) discoidin domain receptor-1 (DDR1) (*Khoshnoodi et al., 2008*; *Brown et al., 2017*). Integrin and DDR1 interactions with type IV collagen mediate a wide variety of cell-matrix functions, including cell migration and cell adhesion (*Khoshnoodi et al., 2008*; *Castro-Sanchez et al., 2010*; *Borza and Pozzi, 2014*; *Xiao et al., 2015*). DDRs and integrin also functionally interact within cells (*Leitinger, 2014*); however, the roles of these interactions in native tissue settings remains poorly defined.

At tissue interfaces juxtaposed BMs usually slide along each other and maintain distinct tissue boundaries (*Sherwood and Sternberg, 2003*; *Brown, 2011*). At specific sites, however, the BMs of neighboring tissues link to stabilize tissue interactions and build complex organs (*Keeley and Sherwood, 2019*). Examples include the kidney, where podocytes and endothelial BMs connect to form the glomerular BM blood filtration unit (*Abrahamson, 1985*); the brain, where the astrocyte end feet and endothelial BMs join to build the blood-brain barrier (*Sixt et al., 2001*); and the somite, where somite and epidermal BMs link to maintain somite-epidermal association during development (*Feitosa et al., 2012*). Approximately 20 different BM-BM linkages between neighboring tissues have been documented (*Gao et al., 2017*; *Keeley and Sherwood, 2019*; *Welcker et al., 2021*). Disruption of these BM-BM tissue connections appears to underlie human diseases, such as in Alport's syndrome, where defects in kidney and hearing are associated with tissue linkage disruption (*Merchant et al., 2004*; *Naylor et al., 2021*). Despite the prevalence and importance of BM-BM tissue connections, the challenge of studying tissue interactions in vivo has hampered our understanding of the mechanisms establishing and maintaining these linkages.

The simple tissues, visual transparency, and experimental tractability of *Caenorhabditis elegans* offer a powerful model to study BM-BM tissue interactions. Further, most *C. elegans* BM components and receptors are endogenously tagged with genetically encoded fluorophores to visualize localization, levels, and dynamics (*Keeley et al., 2020*; *Jayadev et al., 2022*). A BM-BM tissue connection between the large, multinucleated uterine utse cell and epidermal seam cells stabilizes the uterus during egg-laying. The utse-seam connection is formed by BMs of the utse and the seam cells, each ~50 nm thick, which are bridged by an ~100 nm connecting matrix (*Vogel and Hedgecock, 2001*; *Morrissey et al., 2014*; *Gianakas et al., 2023*). Disruption of the utse-seam BM-BM tissue linkage results in prolapse of the uterus (**Rup**tured phenotype) out of the animal after the onset of egg-laying. Formation of the utse-seam attachment is initiated by the deposition of the matrix proteins fibulin and hemicentin during the last larval stage of development. Hemicentin promotes the recruitment of type IV collagen, which accumulates at high levels at the BM-BM tissue connection and strengthens the adhesion, allowing it to resist the strong mechanical forces of egg-laying. The utse-seam connection is robust, with each component of the tissue-spanning matrix contributing to the BM-BM connection (*Gianakas et al., 2023*). This likely accounts for the ability of the utse-seam connection to initially resist mechanical forces after loss of any of these components, delaying the uterine prolapse phenotype until sometime after the initiation of egg-laying. The integrin, αINA-1, one of two *C. elegans* α-integrins, is also expressed in the utse and is required to prevent uterine-prolapse during egg-laying. However, its regulation and function at the utse-seam connection is unclear (*Morrissey et al., 2014*; *Gianakas et al., 2023*). Mechanisms that link juxtaposed BMs appear to be shared, as hemicentin-1 promotes BM-BM tissue connection in the zebrafish fin fold, hemicentin-2 and fibulin-1 mediate zebrafish somite-epidermis BM-BM association, and hemicentin-1 stabilizes BM-BM linkage at myotendinous junctions in mice (*Carney et al., 2010*; *Feitosa et al., 2012*; *Welcker et al., 2021*). In addition, type IV collagen bridges the podocyte BM and endothelial BM to form the glomerular BM in mice and humans (*Suleiman et al., 2013*; *Naylor et al., 2021*). While significant insights are being gained on matrix composition at BM-BM tissue connections, it remains unknown how tissues coordinate and strengthen their adhesive activity at linkage sites. Cells at tissue connections must resist the forces of tissue shifting arising from growth, muscle contraction, and blood flow, and it is unclear how tissues synchronize and bolster adhesions at tissue linkage sites.

Here, we discover that the *C. elegans* DDR-2, an ortholog to the two vertebrate collagen binding DDR RTKs, coordinates and strengthens an integrin-mediated cell adhesion to support the utse-seam BM-BM tissue linkage during its formation. Using genetics, endogenous tagging, and tissue-specific molecular perturbations, we show that DDR-2 is expressed and functions in the utse and seam cells to mediate utse-seam attachment between the mid and late L4 larval stages. We reveal that assembly of type IV collagen linking the juxtaposed BMs triggers the internalization of DDR-2 in both the utse

and seam cells into endocytic vesicles, a signaling compartment for DDR/RTK receptors. In the *ddr-2* knockout mutant, gaps in utse-seam attachment occurred at this time. Through tissue-specific perturbation, quantitative fluorescence analysis, and fluorescence recovery after photobleaching (FRAP), we provide evidence that DDR-2 signals through LET-60/Ras to increase the levels and stability of the α-integrin INA-1 at the utse-seam linkage. Together, these results reveal a mechanism for synchronizing a robust adhesion during tissue linkage, where type IV collagen, which bridges and fastens the BM-BM tissue linkage, also serves as a signal to coordinate and bolster cell-matrix adhesion in cells on both sides of the tissue connection.

## Results

### The utse and seam cells associate during the early L4 stage

The utse is an H-shaped syncytial uterine cell that underlies the *C. elegans* uterus and opens along the central crossbar region during egg-laying to allow embryo passage (*Ghosh and Sternberg, 2014*). The utse is flanked laterally on both sides of the animal by a string of epidermal seam cells that run the length of the body. The utse cell and seam cells are each encased in a BM, which become linked by a specialized matrix whose known functional components include hemicentin, fibulin, and type IV collagen (*Figure 1A*; *Gianakas et al., 2023*). The BM-BM tissue connection supports the uterus during egg-laying and disruption of the linkage results in uterine prolapse (*Figure 1B and C*; *Vogel and Hedgecock, 2001*). The utse and seam cells are in contact by the mid L4 larval stage (*Gianakas et al., 2023*), but when the initial contact is established is not known.

The utse is formed from the fusion of one specialized uterine cell, the anchor cell, with eight adjacent uterine pi cell progeny, forming the syncytial nine-nuclei utse during the early L4 stage (*Newman et al., 1996*; *Ghosh and Sternberg, 2014*). To determine when the utse first contacts the seam cells, we used an anchor cell-specific mCherry fluorophore marker (*cdh-3p::mCherry::PLCδ$^{PH}$*; *Figure 1D*, arrow) (*Hagedorn et al., 2009*) that diffuses into the uterine pi cell progeny during fusion and simultaneously viewed a GFP seam cell marker (*scmp::GFP::CAAX*) (*Chapman et al., 2008*). We found that the utse contacts the seam cells during the early L4 stage as the utse syncytium forms (*Figure 1D*, arrowheads; n=5/5 cases observed). Shortly after utse-seam contact, the BM-BM linkage is initiated at the mid L4 stage by the assembly of low levels of hemicentin, which is secreted by the utse and fibulin-1, which hemicentin recruits from the extracellular fluid. Fibulin-1 and hemicentin protect the utse-seam cell connection from mechanical forces arising from body movement and uterine and vulval muscle contractions prior to egg-laying (*Gianakas et al., 2023*). Hemicentin also promotes recruitment of type IV collagen from the extracellular fluid, which allows the utse-seam BM-BM connection to resist the strong mechanical forces from egg-laying (*Gianakas et al., 2023*; *Figure 1D*). Together, these results show that the utse and seam cells are in contact prior to deposition of matrix components that strengthen the utse-seam BM-BM tissue connection.

### Loss of DDR-2 causes uterine prolapse

As type IV collagen is critical to the BM-BM connection, we were next interested in understanding how receptors for type IV collagen regulate the utse-seam BM-BM linkage. The two main receptors for type IV collagen are integrin and the DDR (*Fidler et al., 2018*). *C. elegans* harbor two integrin heterodimers composed of different α chains with a shared β chain: αINA-1/ βPAT-3 and αPAT-2/ βPAT-3 (*Clay and Sherwood, 2015*). The integrin INA-1 is expressed in the utse and its loss leads to uterine prolapse, suggesting a possible role in utse-seam attachment (*Morrissey et al., 2014*; *Gianakas et al., 2023*). Whether DDRs function at the utse-seam connection, however, is unknown.

*C. elegans* have two DDR genes, *ddr-1* and *ddr-2,* which are paralogs and orthologs to the two mammalian DDRs (*Vogel et al., 2006*). Like mammalian counterparts, the *C. elegans* DDRs are comprised of discoidin and discoidin-like domains, followed by a transmembrane domain and a kinase domain (*Figure 2A*). To determine if *C. elegans* DDRs play a role in utse-seam BM-BM linkage, we examined several *ddr-1* and *ddr-2* mutants (*Figure 2A*). We first screened for the Rup phenotype caused by uterine prolapse, observing animals every day during the egg-laying period, from its onset (48 hr post-L1) to end (120 hr) (Materials and methods). Animals harboring the *ddr-1(ok874)* deletion allele, which eliminates the intracellular portion of DDR-1, and animals with the *ddr-1(tm382)* allele containing an early stop codon truncating the majority of the predicted protein (*Unsoeld et al.,*

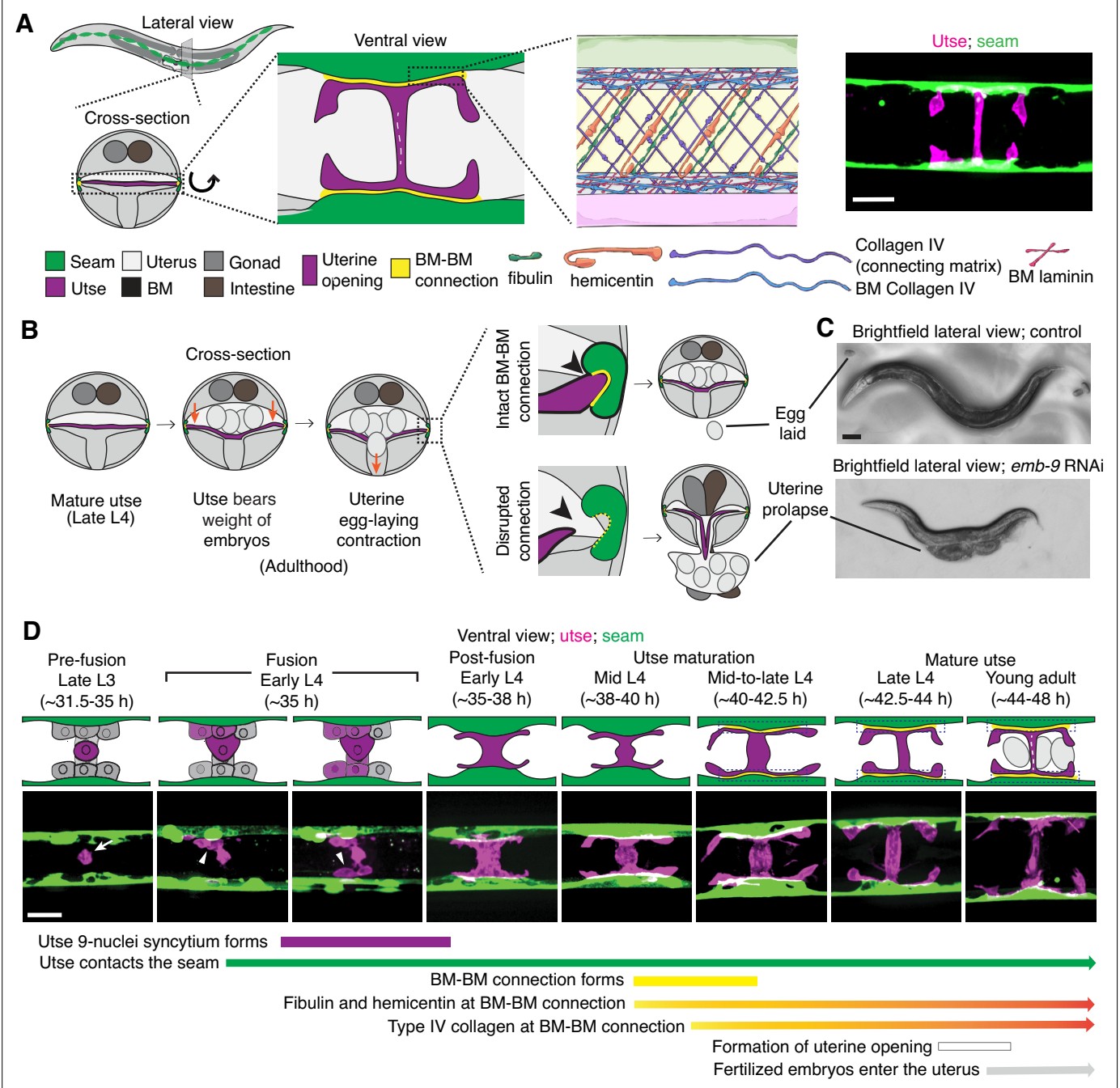

**Figure 1.** Morphogenesis of the *C. elegans* utse-seam basement membrane (BM)-BM tissue connection. (**A**) Left and center: Schematic depicting the utse-seam connection. The utse (a syncytial uterine cell) and the seam epidermal cells are both encased in BMs, which are linked by a BM-BM connecting matrix. Right: Ventral fluorescence z-projection showing the utse and seam cells visualized by the markers *cdh-3p::mCherry::PLC* $^{δPH}$ and *scmp::GFP::CAAX*, respectively. (**B**) Schematic illustrating the role of the utse-seam connection in supporting the uterus during egg-laying muscle contractions. Disruption of the connection results in uterine prolapse. (**C**) Lateral brightfield images of adult worms on control RNAi treatment or RNAi against a key component of the BM-BM linkage, α1 type IV collagen/*emb-9* (RNAi fed from the L1 onward); note the uterine prolapse after loss of collagen. (**D**) A schematic summarizing the development of the utse-seam connection. Fluorescence images shown are ventral z-projections of the utse and the seam at the respective developmental stages from the late L3 to young adult (hours post-hatch at 20°C). Arrow indicates the uterine anchor cell (AC) that fuses with adjacent uterine cells to form the utse. During syncytium formation, the utse makes contact with the seam cells (arrowheads). Dotted boxes denote the BM-BM connection. Scale bars, 20 μm.

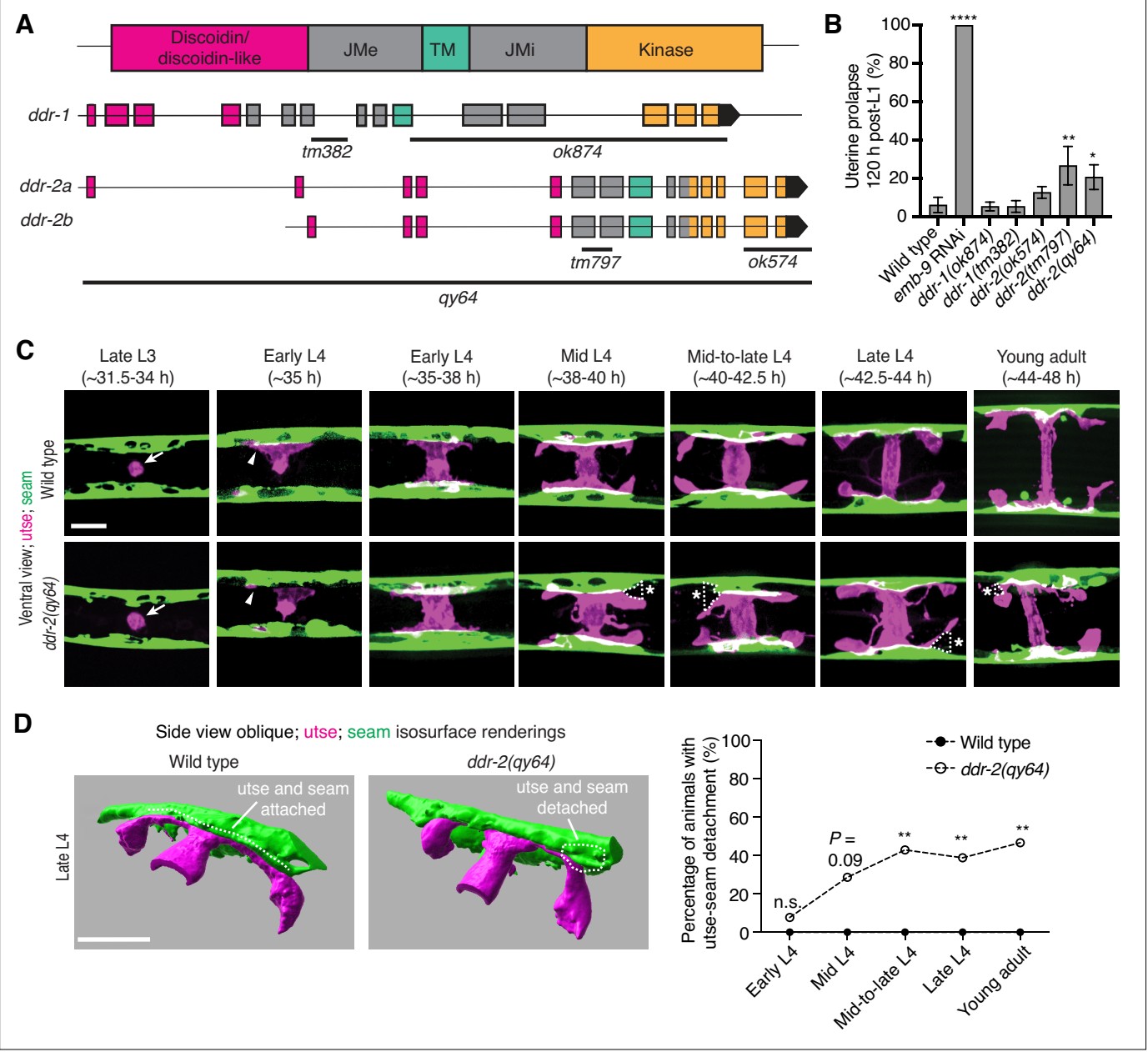

**Figure 2.** Discoidin domain receptor-2 (DDR-2) promotes utse-seam attachment and protects against uterine prolapse. (**A**) Domain structure of *C. elegans* DDRs and exon-intron gene structures of *ddr-1* and *ddr-2*. Exons are color coded according to respective domains. Gene regions that are deleted in respective mutant alleles (italicized text) are demarcated by solid black lines. (**B**) Frequency of uterine prolapse in *ddr-1* and *ddr-2* mutant animals compared to wild type 120 hr post-L1 plating. Data are shown as mean prolapse percentage ± SD, derived from three independent trials (n=50 animals screened per trial). α1 type IV collagen/*emb-9* RNAi was used as a positive control. ****p≤0.0001, **p≤0.01, *p≤0.05; one-way ANOVA with post hoc Dunnett's test. (**C**) Ventral fluorescence z-projections of the utse (*cdh-3p::mCherry::PLC* $^{δPH}$) and seam (*scmp::GFP::CAAX*) cells in wild-type and *ddr-2* knockout (*ddr-2(qy64)*) animals from the late L3 to young adult developmental stages (hours post-hatch at 20°C). Arrows indicate the AC prior to utse formation and arrowheads denote early contacts between the utse and seam. Dotted lines with asterisks indicate regions of utse-seam detachment in *ddr-2(qy64)* animals. (**D**) Left: 3D isosurface renderings of the utse and seam in representative late L4 wild-type and *ddr-2(qy64)* animals (see also *Figure 2—video 1*). Right: Quantification of the percentage of animals with utse-seam detachment. Wild type, early L4 to young adult: n=0/13, n=0/14, n=0/22, n=0/21, n=0/18 animals with detachments respectively; *ddr-2(qy64)*, early L4 to young adult: n=1/13, n=4/14, n=9/22, n=7/18, n=7/15 animals with detachments respectively. **p≤0.01, n.s. (not significant), p>0.05; Fisher's exact test. Scale bars, 20 μm.

The online version of this article includes the following video and source data for figure 2:

**Source data 1.** Source data for *Figure 2*.

**Figure 2—video 1.** Animation of 3D isosurface rendering of the utse and seam in a representative late L4 wild-type animal; related to *Figure 2D*.
https://elifesciences.org/articles/87037/figures#fig2video1

*2013*), did not show a Rup phenotype (*Figure 2A and B*). In contrast, animals homozygous for the *ddr-2(tm797)* allele, which introduces an early stop codon and is thought to be a null or strong loss-of-function (*Unsoeld et al., 2013*), exhibited a significant Rup/uterine prolapse defect. Animals carrying the *ddr-2(ok574)* allele, which deletes a portion of the intracellular kinase domain (*Unsoeld et al., 2013*), also showed an increased frequency of the Rup phenotype compared to wild-type animals, although this difference was not statistically significant (*Figure 2A and B*). We further generated a full-length *ddr-2* deletion allele, *ddr-2(qy64)*, and confirmed that complete loss of *ddr-2* led to a significant uterine prolapse defect (*Figure 2A and B*). Taken together, these results indicate that DDR-2 protects against uterine prolapse and may play a role in mediating utse-seam BM-BM attachment.

## DDR-2 maintains utse-seam attachment during formation of the tissue connection

To determine if the Rup phenotype observed in *ddr-2* mutants is a result of perturbations in the utse-seam BM-BM tissue connection, we examined utse-seam association from the late L3 through young adult stages in animals harboring the *ddr-2(qy64)* deletion allele (*Figure 2C*). We visualized utse-seam association using the utse and seam cell membrane markers *cdh-3p::mCherry::PLCδ^{PH}* and *scmp::GFP::CAAX*, respectively. Animals homozygous for *ddr-2(qy64)* had normal utse-seam cell association at the early L4 stage; however, beginning at the mid L4 stage, gaps in the utse-seam attachment were detected in about 30% of animals. The frequency of gaps increased to ~40% of animals by the mid-to-late L4 and persisted at a similar penetrance to young adulthood (*Figure 2C and D*). In addition, membrane projections emanating from the central body of the utse were detected in *ddr-2(qy64)* animals. These projections were first observed at the mid L4 stage and persisted to young adulthood (*Figure 2C*). These observations suggest that DDR-2 functions around the mid L4 to late L4 stages to promote utse-seam attachment, and that DDR-2 may also regulate utse morphology. Next, we wanted to determine if the utse-seam detachment observed in *ddr-2(qy64)* mutants led to uterine prolapse. Wild-type and *ddr-2(qy64)* animals were mounted and imaged at the L4 larval stage for utse-seam attachment defects, recovered, and tracked to the 72 hr adult stage, where they were examined for the Rup phenotype. All wild-type L4 larvae had intact utse-seam attachment and did not rupture by 72 hr adulthood (n=8/8 animals). However, among nine *ddr-2(qy64)* mutant animals examined, four exhibited utse-seam detachments at the L4 stage, all of which underwent uterine prolapse (Rup) by adulthood. The remaining five *ddr-2* mutant animals, which had an intact utse-seam linkage, did not display the Rup phenotype. Taken together, these results suggest that DDR-2 promotes utse-seam attachment during early stages of tissue linkage.

## DDR-2 is found within endocytic vesicles in the utse and seam cells during initial stages of tissue attachment

As utse-seam detachment in *ddr-2* mutant animals was first observed at the mid L4 larval stage, we next sought to determine if DDR-2 was expressed in the utse, seam, or both cell types at this time. Between the mid L4 and late L4 stage, the utse-seam tissue connection forms into a tongue-and-groove association between the utse (tongue) and seam (groove) (*Figure 2D*, *Figure 2—video 1*). To resolve whether DDR-2 is localized within the seam, utse, or both (*Vogel and Hedgecock, 2001*), we used either a utse (*cdh-3p::mCherry::PLCδ^{PH}*) or seam cell marker (*scmp::2xmKate2::PLCδ^{PH}*), and imaged endogenously tagged DDR-2 protein (DDR-2::mNG) (*Keeley et al., 2020*) in lateral confocal sections from the L4 to young adult stages (*Figure 3A*). We first examined the seam cells contacting the utse (*Figure 3B*). At the early L4 stage DDR-2::mNG was faintly visible at the surface of the seam cells and found in a few intracellular punctae (*Figure 3B*, n=10/10 animals). By the mid L4 stage, DDR-2 was predominantly found in punctae in the region of attachment with the utse. Levels of punctae increased until the late L4 stage and then declined sharply by the young adult (*Figure 3B*). Notably, at the late L4, when DDR-2 punctate localization was at its highest, we could not detect any DDR-2 signal at the cell surface (*Figure 3B*, n=9/9). We further examined DDR-2::mNG in seam cells that were not contacting the utse and found that DDR-2 levels were uniformly low from the early L4 through the young adult stages (*Figure 3—figure supplement 1A*). We next investigated DDR-2 localization in the utse contacting the seam. Like the seam cells at the tissue connection site, DDR-2 in the utse was punctate and peaked in levels between the mid and late L4 stages and then declined rapidly (*Figure 3C*). We conclude that DDR-2 is present at high levels and localized within punctae

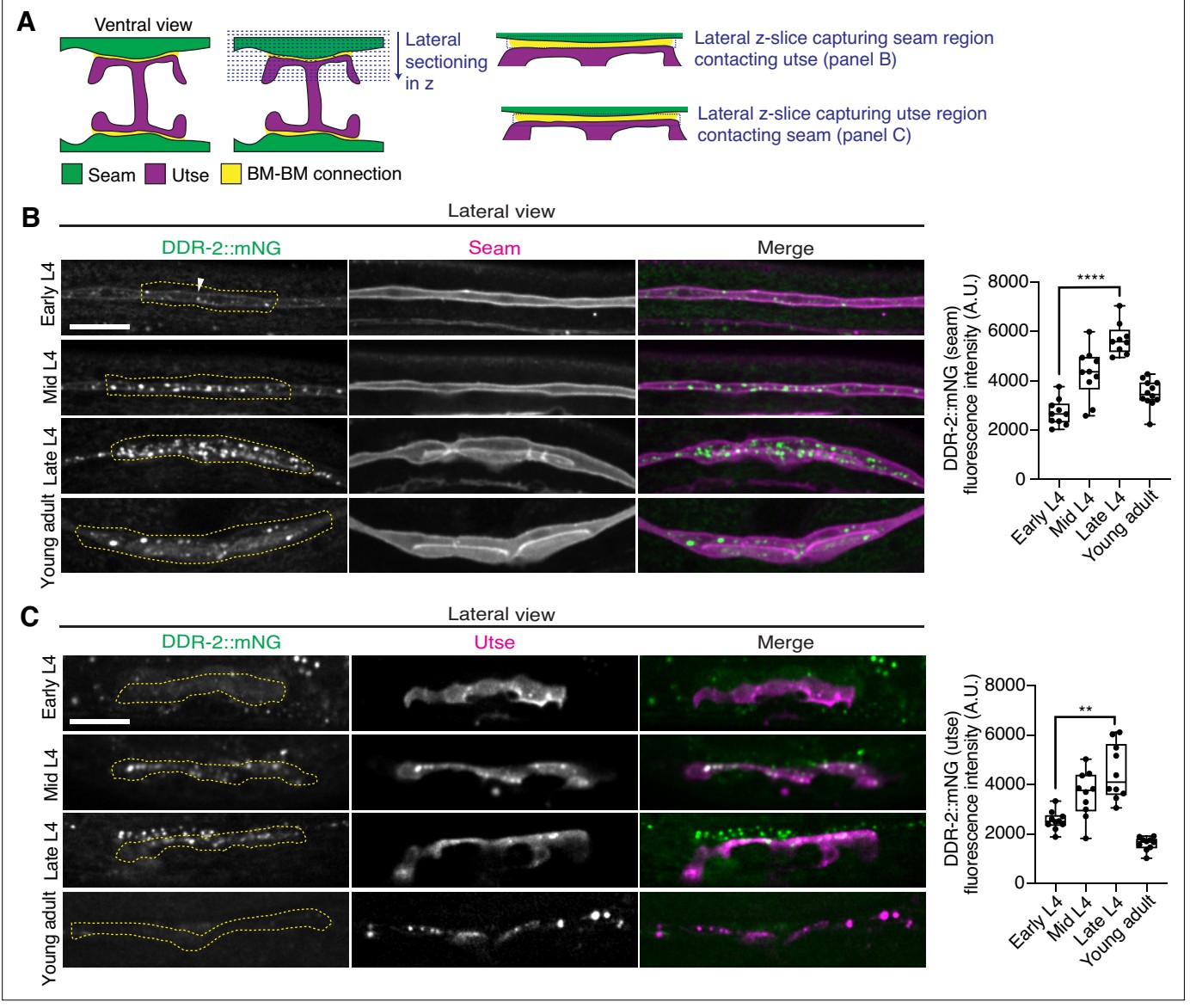

**Figure 3.** Discoidin domain receptor-2 (DDR-2) levels peak in the utse and seam during the formation of the basement membrane (BM)-BM connection. (**A**) Schematic illustrating lateral z-sectioning to determine slices that capture the seam region in contact with the utse and vice versa. (**B**) Left: Lateral fluorescence images of DDR-2::mNG visualized with the seam marker *scmp::2xmKate2::PLC $^{\delta PH}$* from the early L4 to young adult stages. Dotted yellow regions indicate DDR-2 signal in the seam contacting the utse. White arrowhead denotes DDR-2 signal at the cell surface. Right: Quantification of mean DDR-2::mNG fluorescence intensity in the dotted regions (n≥9 for each developmental stage). ****p≤0.0001, one-way ANOVA with post hoc Dunnett's test. (**C**) Left: Lateral fluorescence images of DDR-2::mNG visualized with the utse marker *cdh-3p::mCherry::PLC $^{\delta PH}$* from the early L4 to young adult stages. Dotted yellow regions indicate DDR-2 signal in the utse contacting the seam. Right: Quantification of mean DDR-2::mNG fluorescence intensity in dotted yellow regions (n=10 all stages). **p≤0.01, one-way ANOVA with post hoc Dunnett's test. Scale bars, 20 µm. Box edges in boxplots represent the 25th and 75th percentiles, the line in the box denotes the median value, and whiskers mark the minimum and maximum values.

The online version of this article includes the following source data and figure supplement(s) for figure 3:

**Source data 1.** Source data for *Figure 3*.

**Figure supplement 1.** Discoidin domain receptor-2 (DDR-2) is found within endocytic vesicles.

**Figure supplement 1—source data 1.** Source data for *Figure 3—figure supplement 1*.

structures in the seam and utse cells at the time when utse-seam defects are observed in *ddr-2* mutant animals.

RTKs, including DDRs, are activated upon ligand binding and then rapidly internalized into endocytic vesicles: first the early endosome (marked by RAB-5), followed by subsequent sorting

for either lysosome-mediated degradation via late endosomes (marked by RAB-7) or recycled back to the cell surface in recycling endosomes (marked by RAB-11) (*Goh and Sorkin, 2013*). In vertebrates, the exposure of cells to collagen results in the internalization of DDR1b into Rab5a-positive early endosomes (*Mihai et al., 2009*). Phosphorylation of DDR1 does not occur until the onset of endocytosis and recycling, suggesting that receptor activation occurs in endocytic vesicles (*Mihai et al., 2009*; *Fu et al., 2013*). As *C. elegans* DDR-2 appeared to be functioning during the mid L4 to late L4 stages, when levels of DDR-2 were highest and present in vesicular structures, we sought to determine whether DDR-2 was internalized within endocytic vesicles. We examined localization of DDR-2::mNG punctae in relation to utse-expressed markers of early endosomes (*cdh-3p::mCherry::rab-5*), late endosomes (*cdh-3p::mCherry::rab-7*), and recycling endosomes (*cdh-3p::mCherry::rab-11*) (*Sato et al., 2014*). DDR-2 was co-localized with all three classes of endosomes in the utse, with the highest overlap with RAB-5 early endosomes (*Figure 3—figure supplement 1B*). In addition, we examined an early endosome marker expressed in seam cells (*scmp::mKate2::rab-5*) and observed DDR-2::mNG punctae co-localized with RAB-5 vesicles (*Figure 3—figure supplement 1C*). The presence of DDR-2 in endocytic vesicles within the utse and seam suggested that DDR-2 signaling may be required for utse-seam attachment. Consistent with this, animals harboring the *ddr-2(ok574)* mutant allele, where a portion of the intracellular kinase domain is deleted (*Figure 2A*), exhibited significant utse-seam detachments at the late L4 stage (*Figure 4A*). Taken together, these observations suggest that DDR-2 functions in the utse and seam cells to promote attachment at the tissue linkage site between the mid and late L4 stages, where it may signal within endocytic vesicles.

## Type IV collagen promotes DDR-2 vesiculation

DDR signaling is triggered by collagen binding, and vertebrate DDRs bind multiple collagens with different binding affinities (*Leitinger, 2014*). Genetic interaction studies in *C. elegans* revealed that transmembrane collagen COL-99, which is most similar to transmembrane collagen type XIII, XXIII, and XXV in vertebrates (*Tu et al., 2015*), may act as a ligand for *C. elegans* DDR-1 and DDR-2 during axon guidance and that type IV collagen might act as a ligand for DDR-2 during axon regeneration (*Hisamoto et al., 2016*; *Taylor et al., 2018*). *C. elegans* BMs also harbor type XVIII collagen (CLE-1) (*Keeley et al., 2020*). Thus, we next investigated whether any of these cell and BM-associated collagens could function as a ligand for DDR-2 at the utse-seam BM-BM tissue connection site.

Type IV collagen enriches at the utse-seam BM-BM linkage during its maturation and helps the connected tissues resist the mechanical forces of egg-laying. In contrast, type XVIII collagen is not enriched at the linkage (*Gianakas et al., 2023*). Whether COL-99 localizes to the utse-seam tissue linkage is unknown. We thus first examined an endogenously tagged COL-99 (mNG::COL-99) (*Jayadev et al., 2022*) and compared its localization and levels to type IV collagen (α1-type IV collagen::mNG (EMB-9::mNG)) and type XVIII collagen (CLE-1::mNG) (*Keeley et al., 2020*) at the utse-seam connection in late L4 animals. COL-99 was not detected while EMB-9 was present at nearly eightfold higher levels than CLE-1, suggesting that type IV collagen might be the ligand for DDR-2 (*Figure 4B*). Furthermore, the enrichment of type IV collagen levels at the utse-seam connection site occurs rapidly from the mid L4 to late L4 stage (*Gianakas et al., 2023*), correlating with the buildup of DDR-2 in endosomes in the utse and seam cells.

As exposure to collagen triggers vesiculation of vertebrate DDR1b into endocytic vesicles, we wanted to determine whether type IV collagen controls vesiculation of DDR-2 at the utse-seam BM-BM connection. RNAi-mediated depletion of type IV collagen from the L1 through the late L4 larval stages resulted in a sharp decrease in vesicular DDR-2::mNG in the utse and seam cells at the linkage site, accompanied by enrichment of DDR-2 at cell surfaces (*Figure 4C*; *Figure 4—figure supplement 1A*). In contrast, reduction of COL-99 and type XVIII collagen did not significantly affect DDR-2 vesiculation (*Figure 4C*; *Figure 4—figure supplement 1B and C*). Depletion of hemicentin, which plays a crucial role in recruiting type IV collagen to the BM-BM linking matrix (*Gianakas et al., 2023*), also resulted in a significant reduction in DDR-2 vesiculation (*Figure 4D*; *Figure 4—figure supplement 1D*). In addition, we did not detect DDR-2 at the cell surface, suggesting that hemicentin has a role in recruiting DDR-2 to the site of utse-seam attachment. It is possible that collagen could also function in DDR-2 recruitment, but we could not assess this definitively due to the lower knockdown efficiency of *emb-9* RNAi (*Figure 4—figure supplement 1A*). Together, these findings indicate that type IV

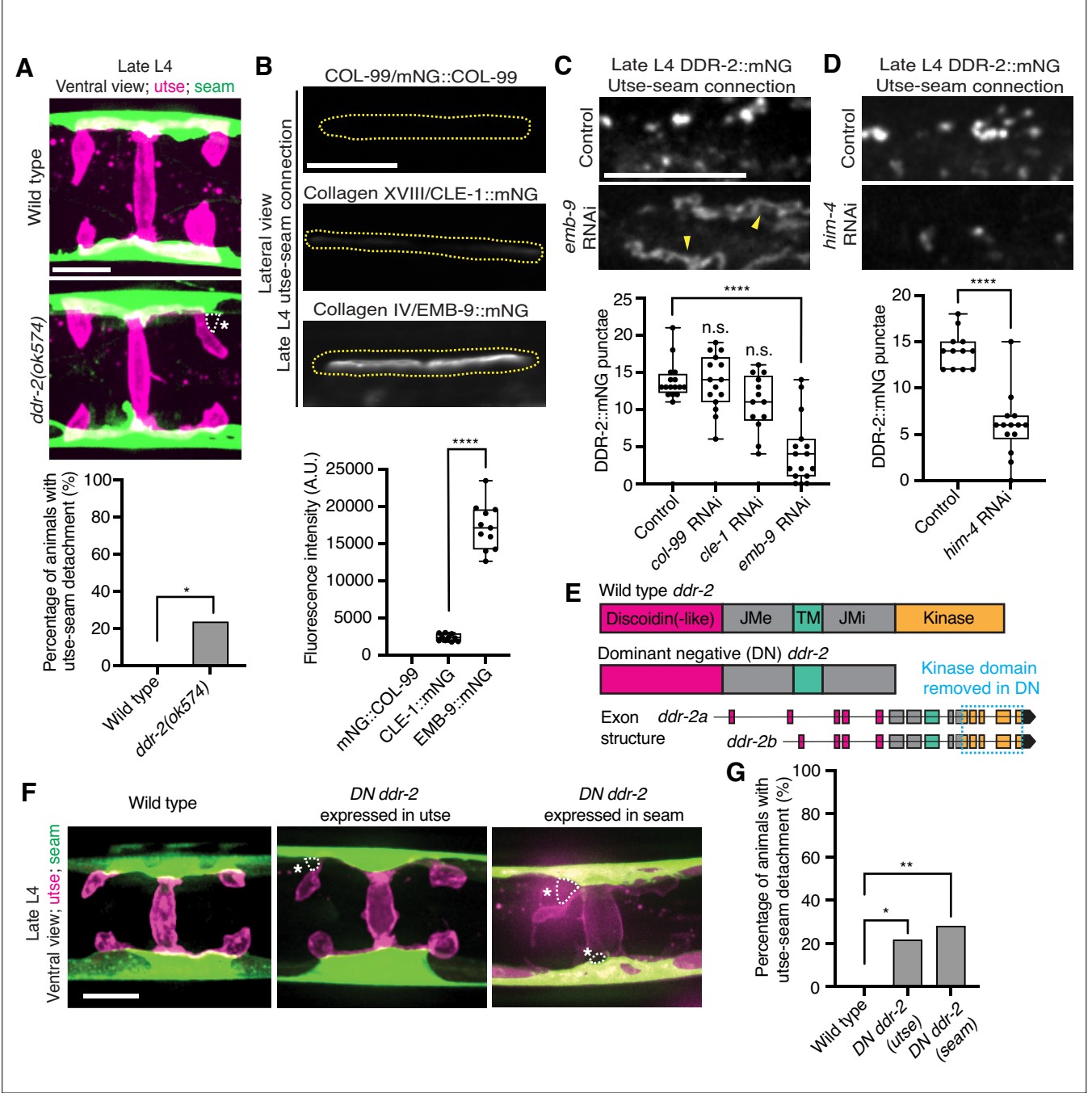

**Figure 4.** Discoidin domain receptor-2 (DDR-2) endocytosis is triggered by type IV collagen and DDR-2 is required in both the utse and seam to promote tissue connection. (**A**) Top: Ventral fluorescence z-projections of the utse (*nas-22p::2xmKate2::PLC$^{δPH}$*) and seam (*scmp::GFP::CAAX*) cells in late L4 wild-type and *ddr-2* kinase domain truncation mutant (*ddr-2(ok574)*) animals. Dotted lines with asterisks indicate regions of utse-seam detachment. Bottom: Quantification of the percentage of animals with utse-seam detachment. Wild type, n=0/26; *ddr-2(ok574)*, n=5/21 animals with detachments respectively. **p≤0.01, Fisher's exact test. (**B**) Top: Lateral fluorescence images of COL-99 (mNG::COL-99), type XVIII collagen (CLE-1::mNG), and α1 type IV collagen (EMB-9::mNG) at the utse-seam connection (dotted yellow regions) of late L4 animals. All images were acquired at the same exposure. Bottom: Quantification of mean fluorescence intensity (n≥10 for each fluorescent protein). ****p≤0.0001, unpaired two-tailed Student's *t* test. (**C**) Top: Lateral fluorescence z-projections of DDR-2::mNG at the utse-seam connection in control or *emb-9* RNAi-treated late L4 animals. Note that the field of view contains a single utse arm contacting the seam tissue. Arrowheads indicate cell surface accumulation of DDR-2 upon collagen IV depletion. Bottom: Quantification of the average number of discrete DDR-2 punctae at the utse-seam connection on control, *col-99*, *cle-1*, or *emb-9* RNAi treatments (n≥13 animals for all treatments). ****p≤0.0001, n.s. (not significant), p>0.05; Kruskal-Wallis *H* test with post hoc Dunn's test. (**D**) Top: Lateral fluorescence z-projections of DDR-2::mNG at the utse-seam attachment in control or hemicentin/*him-4* RNAi-treated late L4 animals. Bottom:

*Figure 4 continued on next page*

*Figure 4 continued*

Quantification of the average number of discrete DDR-2 punctae at the utse-seam attachment on control or *him-4* RNAi-treated animals (n≥13 all treatments). ****p≤0.0001, Mann-Whitney *U* test. (**E**) Schematic of dominant negative (DN) *ddr-2*. (**F**) Ventral fluorescence z-projections of the utse (*nas-22p::2xmKate2::PLC δPH*) and seam (*wrt-2p::GFP::PLC δPH*) tissues in late L4 stage wild-type animals and animals expressing DN *ddr-2* in the utse (driven by *cdh-3* promoter) or the seam (driven by *scm* promoter). Dotted lines with asterisks indicate regions of utse-seam detachment. (**G**) Quantification of the percentage of animals with utse-seam detachment. Wild type, n=0/23; DN *ddr-2(utse)*, n=5/23; DN *ddr-2(seam)*, n=7/25 animals with detachments respectively. **p≤0.01, *p≤0.05; Fisher's exact test. Scale bars, 20 μm. Box edges in boxplots represent the 25th and 75th percentiles, the line in the box denotes the median value, and whiskers mark the minimum and maximum values.

The online version of this article includes the following source data and figure supplement(s) for figure 4:

**Source data 1.** Source data for *Figure 4*.

**Figure supplement 1.** Collagen and hemicentin knockdown efficiencies.

**Figure supplement 1—source data 1.** Source data for *Figure 4—figure supplement 1*.

**Figure supplement 2.** *ddr-2* loss does not reduce functional levels of fibulin, hemicentin, type IV collagen, and matrix metalloproteinase ZMP-4 at the utse-seam connection.

**Figure supplement 2—source data 1.** Source data for *Figure 4—figure supplement 2*.

collagen assembly at the site of utse-seam BM-BM tissue linkage triggers DDR-2 vesiculation and may act as a ligand for DDR-2 activation to promote utse-seam attachment.

## DDR-2 functions in the utse and the seam to promote utse-seam attachment

Since DDR-2 localizes to both the utse and seam cells, we next investigated where DDR-2 functions to promote utse-seam BM-BM attachment. We generated genome-edited lines with single-copy insertions of dominant negative (DN) *ddr-2* (*Bernadskaya et al., 2019*) driven by the *cdh-3* promoter for utse expression or the *scm* promoter for seam expression. Using the utse-specific membrane marker (*nas-22p::2xmKate2::PLCδPH*) (*Park et al., 2010*) and a seam cell membrane marker (*wrt-2p::GFP::PLCδPH*) (*Wildwater et al., 2011*), we found significant utse-seam detachments in both utse-expressed and seam-expressed DN *ddr-2* animals at the late L4 stage (*Figure 4F and G*). Consistent with the timing and vesicular pattern of DDR-2 localization, these results indicate that DDR-2 functions in both the utse and seam between the mid and late L4 stages to promote utse-seam attachment.

## Loss of DDR-2 increases type IV collagen at the utse-seam BM-BM adhesion site

The matrix components fibulin-1 and hemicentin play early roles in mediating utse-seam attachment and hemicentin promotes recruitment of type IV collagen and further enrichment of fibulin, which strengthens the BM-BM attachment (*Gianakas et al., 2023*). We thus next asked whether DDR-2 promotes utse-seam linkage by regulating the assembly of fibulin, hemicentin, or type IV collagen at the connecting matrix. Knockdown of *ddr-2* by RNAi did not alter fibulin levels at the late L4 stage (*Figure 4—figure supplement 2A and B*). However, we observed a modest ~20% reduction in hemicentin upon depletion of DDR-2 (*Figure 4—figure supplement 2C*). As loss of hemicentin results in utse-seam detachment at the late L4 stage (*Gianakas et al., 2023*), we tested whether an ~20% reduction of hemicentin could cause these defects by initiating RNAi at the early L4, just before the utse forms (Materials and methods). We found that utse-seam attachments were intact in all animals at the late L4 stage upon ~20% reduction of hemicentin (n=13/13 and 12/12 control and *him-4* RNAi-treated animals examined respectively; *Figure 4—figure supplement 2D*), suggesting that the utse-seam detachment defect in *ddr-2* mutant animals is not caused by a modest reduction in hemicentin levels. Interestingly, there was an ~40% increase in type IV collagen levels at the utse-seam connection after RNAi-mediated reduction of DDR-2, indicating a possible feedback mechanism between DDR-2 signaling and collagen assembly (*Figure 4—figure supplement 2E*). In sum, these results indicate that knockdown of *ddr-2* does not decrease the functional levels of key matrix components at the site of utse-seam BM-BM connection, suggesting that DDR-2 does not promote utse-seam attachment by regulating the assembly of the BM-BM linking matrix.

# DDR-2 promotes integrin function to mediate utse-seam tissue attachment

Studies have indicated that vertebrate DDRs are not strong adhesive receptors (*Xu et al., 2012*). Consistent with this, we detected weak DDR-2::mNG fluorescence signals at the cell membranes of the utse and seam. However, we observed strong DDR-2 signals in a vesicular pattern in both tissues. We thus assessed several molecular activities regulated by DDR signaling that might mediate utse-seam attachment. Vertebrate DDRs promote matrix metalloproteinase (MMP) expression and localization (*Leitinger, 2014*). The *C. elegans* genome harbors six MMP genes, named zinc metalloproteinase 1–6 (*zmp-1–6*) (*Altincicek et al., 2010*). We examined four available reporters of ZMP localization (ZMP-1::GFP, ZMP-2::GFP, ZMP-3::GFP, and ZMP-4::GFP) (*Kelley et al., 2019*). Only ZMP-4 was detected at the utse-seam connection and its localization was not altered by knockdown of *ddr-2* (*Figure 4—figure supplement 2F*). These observations suggest that DDR-2 does not promote utse-seam linkage through regulation of MMPs, although we cannot rule out roles for DDR-2 in promoting the expression or localization of ZMP-5 or ZMP-6.

DDRs also modulate integrin activity (*Leitinger, 2014*). Cell culture studies have shown that vertebrate DDRs can enhance integrin adhesion and integrin cell surface levels in multiple cell types (*Xu et al., 2012*; *Staudinger et al., 2013*; *Bayer et al., 2019*). *C. elegans* harbor only two integrin receptors, made up of either the α subunit INA-1 or PAT-2 dimerized to the sole β subunit PAT-3 (*Clay and Sherwood, 2015*). We previously found that depletion of INA-1 results in a Rup phenotype, but the specific effect on utse-seam BM-BM tissue attachment was not determined (*Morrissey et al., 2014*). Thus, we next asked whether DDR-2 promotes utse-seam attachment through regulation of integrin. First, we analyzed the localization and levels of endogenously tagged INA-1 (αINA-1::mNG) and PAT-2 (αPAT-2::2xmNG) (*Jayadev et al., 2019*) and found that both integrin α subunits were expressed in the utse and seam tissues and localized to the utse-seam connection region from the mid-L4 stage through young adulthood (*Figure 5A and B*; *Figure 5—figure supplement 1A and B*). Their levels mirrored that of DDR-2, with a peak at the late L4 stage, followed by a drop-off in the young adult stage (*Figure 5A and B*). However, INA-1 levels were at least sixfold greater than PAT-2 (note, each PAT-2 is linked to two mNG molecules, see Materials and methods), suggesting it may have a dominant role in adhesion (*Figure 5A and B*). Loss of *ddr-2* resulted in a significant reduction in levels of both INA-1 and PAT-2 at the utse-seam BM-BM attachment through the L4 larval stage (*Figure 5B*). Taken together these data suggest that DDR-2 could function to promote integrin adhesion within each tissue at the utse-seam connection.

We focused on a possible role for DDR-2 in regulating INA-1 at the utse-seam linkage, as INA-1 was present at higher levels compared to PAT-2 (*Figure 5A and B*) and was more enriched at cell surfaces (*Figure 5—figure supplement 1C*). Furthermore, PAT-2 anchors muscle attachments (*Moerman and Williams, 2006*; *Gieseler et al., 2017*; *Figure 5A*) and its loss leads to paralysis, which eliminates mechanical stress on the tissue connection and prevents defects in the utse-seam attachment (*Gianakas et al., 2023*). To determine if DDR-2 promotes integrin activity, we reduced INA-1 levels at the late L4 stage to 30% by RNAi and assessed whether loss of *ddr-2* enhanced the Rup and utse-seam splitting defects (*Figure 5—figure supplement 1D and E*). Loss of *ddr-2* strongly enhanced the uterine prolapse defect caused by RNAi-mediated knockdown of *ina-1* (*Figure 5C*). In addition, loss of *ddr-2* enhanced the frequency of utse-seam detachments at the late L4 stage upon depletion of INA-1 (*Figure 5D*). We note that utse-seam detachments were not detected upon RNAi against *ina-1* alone, likely due to incomplete knockdown at this timepoint (*Figure 5—figure supplement 1D and E*). Together, these observations are consistent with a possible role for DDR-2 in regulating integrin function to promote utse-seam connection.

To determine where integrin functions to promote utse-seam attachment, we used a previously characterized DN strategy, where expression of the β integrin PAT-3 lacking the extracellular domain (referred to as *DN integrin*; *Figure 5E*) inhibits endogenous integrin function (*Martin-Bermudo and Brown, 1999*; *Lee et al., 2001*; *Hagedorn et al., 2009*). We used the utse-specific *zmp-1^{mK50-51}* promoter (*Figure 5—figure supplement 1F*; *Hagedorn et al., 2009*) and the seam-specific *wrt-2* promoter to disrupt integrin activity in the respective tissues through the time of utse-seam tissue linkage (late L3 to young adult stages, *Figure 1D*). Expression of *DN integrin* in either the utse or the seam cells resulted in utse-seam detachment at the late L4 (*Figure 5F and G*). Taken together, these findings indicate that DDR-2 may promote utse-seam BM-BM adhesion in part by regulating integrin levels and function.

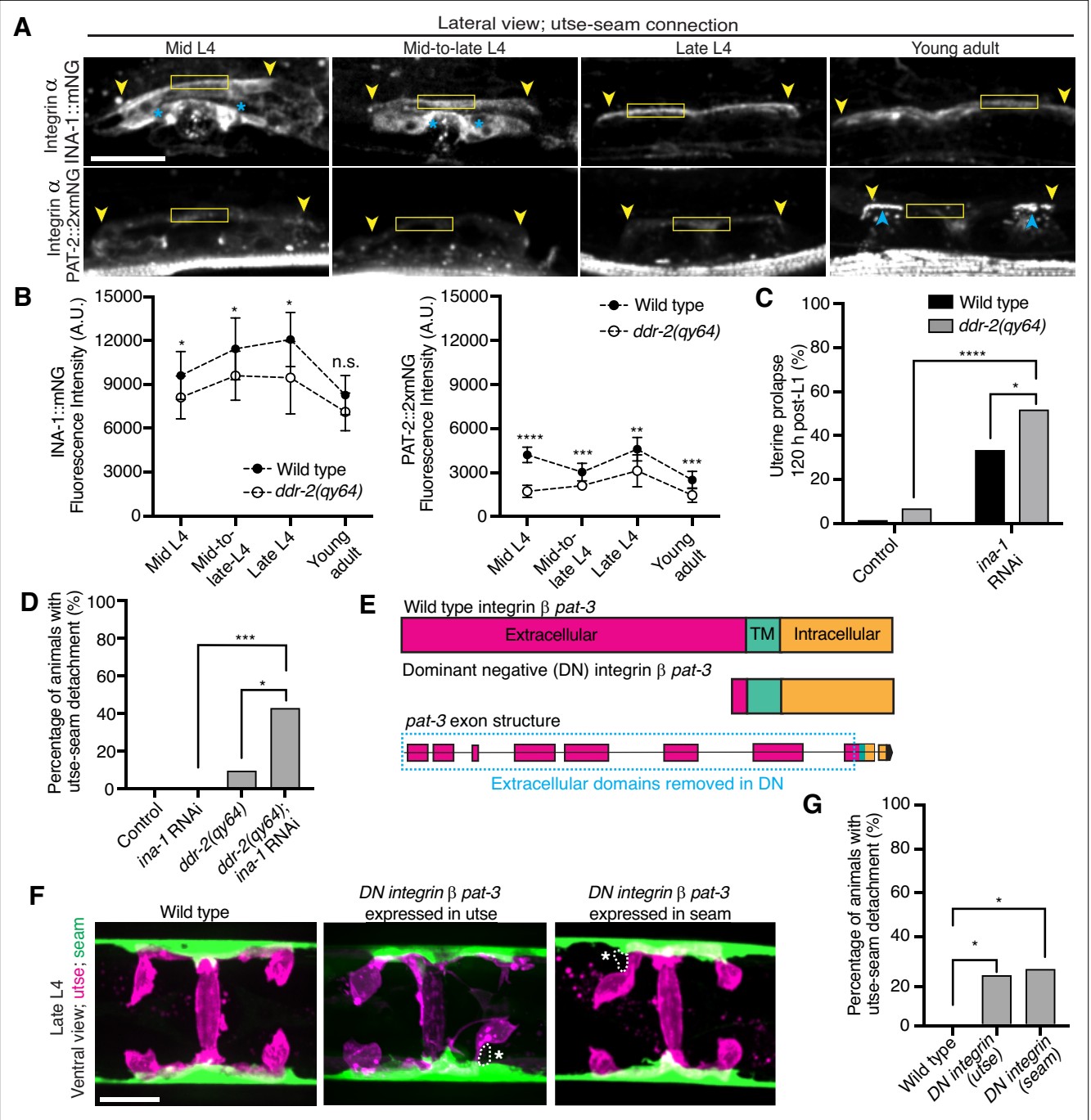

**Figure 5.** Discoidin domain receptor-2 (DDR-2) regulates integrin levels at the utse-seam connection and integrin functions in the utse and seam to mediate tissue attachment. (**A**) Lateral fluorescence z-projections of integrin α subunits INA-1::mNG and PAT-2::2XmNG at the utse-seam connection (yellow arrowheads) from the mid L4 to young adult stages. Blue asterisks and arrowheads denote INA-1 signal in the developing vulva and PAT-2 at muscle attachment sites, respectively. Fluorescence intensity at the utse-seam connection measured in yellow boxes. (**B**) Quantification of mean fluorescence intensity in wild-type and *ddr-2* knockout (*ddr-2(qy64)*) animals (n≥9 all conditions). Error bars represent SD. ****p≤0.0001, ***p≤0.001, **p≤0.01, *p≤0.05, n.s. (not significant), p>0.05; unpaired two-tailed Student's *t* test. (**C**) Frequency of uterine prolapse in wild-type and *ddr-2(qy64)* animals treated with control or *ina-1* RNAi 120 hr post-L1 plating. Control n=1/70, *ddr-2(qy64)* n=4/60, *ina-1* RNAi n=22/66, and *ddr-2(qy64); ina-1* RNAi n=31/60 animals with uterine prolapse respectively. ****p≤0.0001, *p≤0.05; Fisher's exact test. (**D**) Frequency of utse-seam detachments in late L4 wild-type and *ddr-2(qy64)* animals treated with control or *ina-1* RNAi. Control n=0/21, *ina-1* RNAi n=0/25, *ddr-2(qy64)* n=2/21, and *ddr-2(qy64); ina-1* RNAi n=9/21 animals with detachments, respectively. ***p≤0.001, *p≤0.05; Fisher's exact test. (**E**) Schematic of dominant negative (DN) β integrin *pat-3*. (**F**) Ventral fluorescence z-projections of the utse (*cdh-3p::mCherry::PLC* δPH or *nas-22p::2xmKate2::PLC* δPH) and seam (*wrt-2p::GFP::PLC* δPH or *scmp::GFP::CAAX*) cells in late L4 wild-type animals and animals expressing DN integrin in the utse (*zmp-1*mk50-51 promoter) or the seam (*wrt-2* promoter).

*Figure 5 continued on next page*

*Figure 5 continued*

Dotted lines with asterisks indicate regions of utse-seam detachment. (**G**) Percentage of animals with utse-seam detachment. Wild type, n=0/19; *DN integrin (utse)*, n=4/18; *DN integrin (seam)*, n=4/16 animals with detachments, respectively. *p≤0.05, Fisher's exact test. Scale bars, 20 μm.

The online version of this article includes the following source data and figure supplement(s) for figure 5:

**Source data 1.** Source data for *Figure 5*.

**Figure supplement 1.** The integrin α subunits INA-1 and PAT-2 are localized within both the utse and seam cells.

**Figure supplement 1—source data 1.** Source data for *Figure 5—figure supplement 1*.

## DDR-2 promotes vinculin localization and integrin stability at the utse-seam linkage

As DDR-2 promotes integrin localization to the utse-seam BM-BM linkage and integrin activity promotes utse-seam attachment, we further explored how DDR-2 may regulate integrin. Talin and vinculin are core components of the integrin adhesion complex that support integrin activation (*Chastney et al., 2021*). We thus examined endogenously tagged talin (GFP::TLN-1) (*Walser et al., 2017*) and vinculin (DEB-1::mNG) localization and detected both proteins at the utse-seam attachment site. Levels of each protein ramped up to the late L4 and then decreased by the young adult stage, mirroring DDR-2 (*Figure 6A and B*). However, loss of *ddr-2* only decreased DEB-1 levels (*Figure 6B*). Confirming a role in utse-seam connection, depletion of DEB-1 by RNAi resulted in utse-seam detachment at the late L4 stage (*Figure 6C*; *Figure 6—figure supplement 1*). These data show that DDR-2 also regulates vinculin levels at the utse-seam linkage and that vinculin is required for utse-seam attachment.

Integrin adhesions form stable complexes that anchor cells to ECM (*Chastney et al., 2021*). To determine whether DDR-2 regulates integrin complex stability at the site of utse-seam BM-BM tissue connection, we analyzed the dynamics of INA-1 by performing FRAP experiments. We focused on INA-1::mNG at the late L4 stage when DDR-2 peaks in levels. Strikingly, loss of *ddr-2* nearly doubled the rate of recovery of INA-1::mNG fluorescence signal in the bleached region 10 min post-bleaching as compared to wild-type animals (*Figure 6D*). These observations indicate that DDR-2 stabilizes the INA-1 integrin adhesion complex at the utse-seam adhesion site.

## Ras acts in the DDR-2 pathway and promotes integrin adhesion at the utse-seam linkage

Vertebrate DDR2 activates multiple signaling networks, including the PI3K/Akt and Ras/Raf/Erk cascades (*Payne and Huang, 2014*; *Chen et al., 2021*). Notably, Ras regulates integrin activity in several mammalian cell types (*Zhang et al., 1996*; *Conklin et al., 2010*; *Sandri et al., 2012*; *Lilja et al., 2017*). We thus hypothesized that DDR-2 could promote integrin stabilization through Ras signaling. We first strongly depleted LET-60 (see Materials and methods), the *C. elegans* Ras ortholog, and observed penetrant utse-seam attachment defects, which were not worsened in *ddr-2* null mutant animals (*Figure 7A*). This suggests that DDR-2 and LET-60/Ras function in the same pathway to mediate the utse-seam connection. Next, we analyzed INA-1::mNG levels and stability after LET-60 reduction. RNAi against *let-60* resulted in significantly lower INA-1 levels at the utse-seam BM-BM connection site (*Figure 7B*, pre-bleach panels and *Figure 7—figure supplement 1*). Further, like loss of *ddr-2*, FRAP analysis indicated that depletion of LET-60/Ras reduced the stability of INA-1 at the utse-seam linkage site (i.e., there was more rapid recovery of INA-1 after photobleaching; *Figure 7B*). We also generated animals harboring a gain-of-function mutation in Ras (LET-60$^{G13E}$, *let-60(qy203)*) (*Singh and Han, 1995*). Strikingly, utse-seam detachments were observed in *let-60* gain-of-function animals (*Figure 7C*) and FRAP experiments revealed that INA-1 had an approximately twofold greater stability at the utse-seam BM-BM connection site (*Figure 7D*). Collectively, these findings suggest that Ras acts downstream of DDR-2 to set the appropriate levels and stability of INA-1-mediated adhesion during the formation of the utse-seam BM-BM tissue connection.

## Discussion

To form organs such as the brain, kidney, and lung, separate tissues connect through adjoining BMs to mediate complex functions, such as molecular barrier, blood filtration, and gas exchange, respectively (*Keeley and Sherwood, 2019*). These linkages resist the independent movement of each tissue and

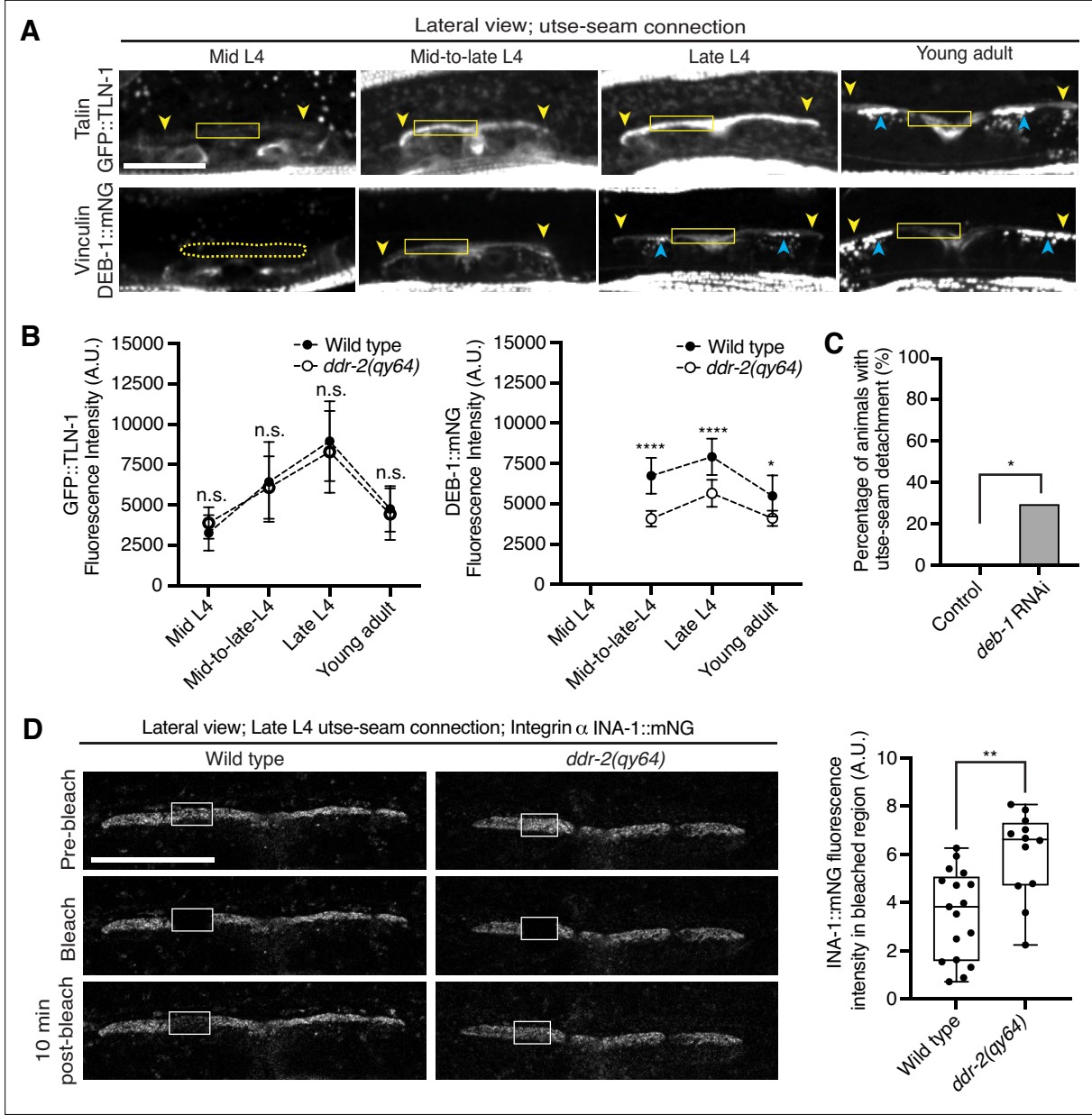

**Figure 6.** Discoidin domain receptor-2 (DDR-2) upregulates vinculin levels and stabilizes integrin at the utse-seam connection. (**A**) Lateral fluorescence z-projections of the integrin activators talin (GFP::TLN-1) and vinculin (DEB-1::mNG) at the utse-seam attachment region (bounded by yellow arrowheads) from the mid L4 to young adult stages. Vinculin was not detected at the utse-seam attachment region at the mid L4 stage (dotted yellow box). Blue arrowheads indicate fluorescence signal at muscle attachment sites. Fluorescence intensity at the utse-seam attachment region was measured in the solid yellow boxes. (**B**) Quantification of TLN-1 and DEB-1 mean fluorescence intensity in wild-type and *ddr-2* knockout (*ddr-2(qy64)*) animals (n≥9 all conditions). Error bars represent SD. ****p≤0.0001, **p≤0.01, *p≤0.05, n.s. (not significant), p>0.05; unpaired two-tailed Student's *t* test. (**C**) Frequency of utse-seam detachments observed in control or *deb-1* RNAi-treated late L4 animals. Control n=0/21, *ina-1* RNAi n=5/17 animals with detachments respectively. *p≤0.05, Fisher's exact test. (**D**) Left: Lateral fluorescence images of INA-1::mNG at the utse-seam connection before photobleaching, immediately after photobleaching, and 10 min post-photobleaching in late L4 wild-type and *ddr-2(qy64)* animals. Box indicates bleached region. Right: Quantification of mean INA-1::mNG fluorescence intensity in the bleached region 10 min post-photobleaching. Wild type, n=17; *ddr-2(qy64)*, n=12. **p≤0.01, unpaired two-tailed Student's *t* test. Scale bars, 20 µm. Box edges in boxplots represent the 25th and 75th percentiles, the line in the box denotes the median value, and whiskers mark the minimum and maximum values.

The online version of this article includes the following source data and figure supplement(s) for figure 6:

**Source data 1.** Source data for *Figure 6*.

**Figure supplement 1.** *deb-1* knockdown efficiency.

**Figure supplement 1—source data 1.** Source data for *Figure 6—figure supplement 1*.

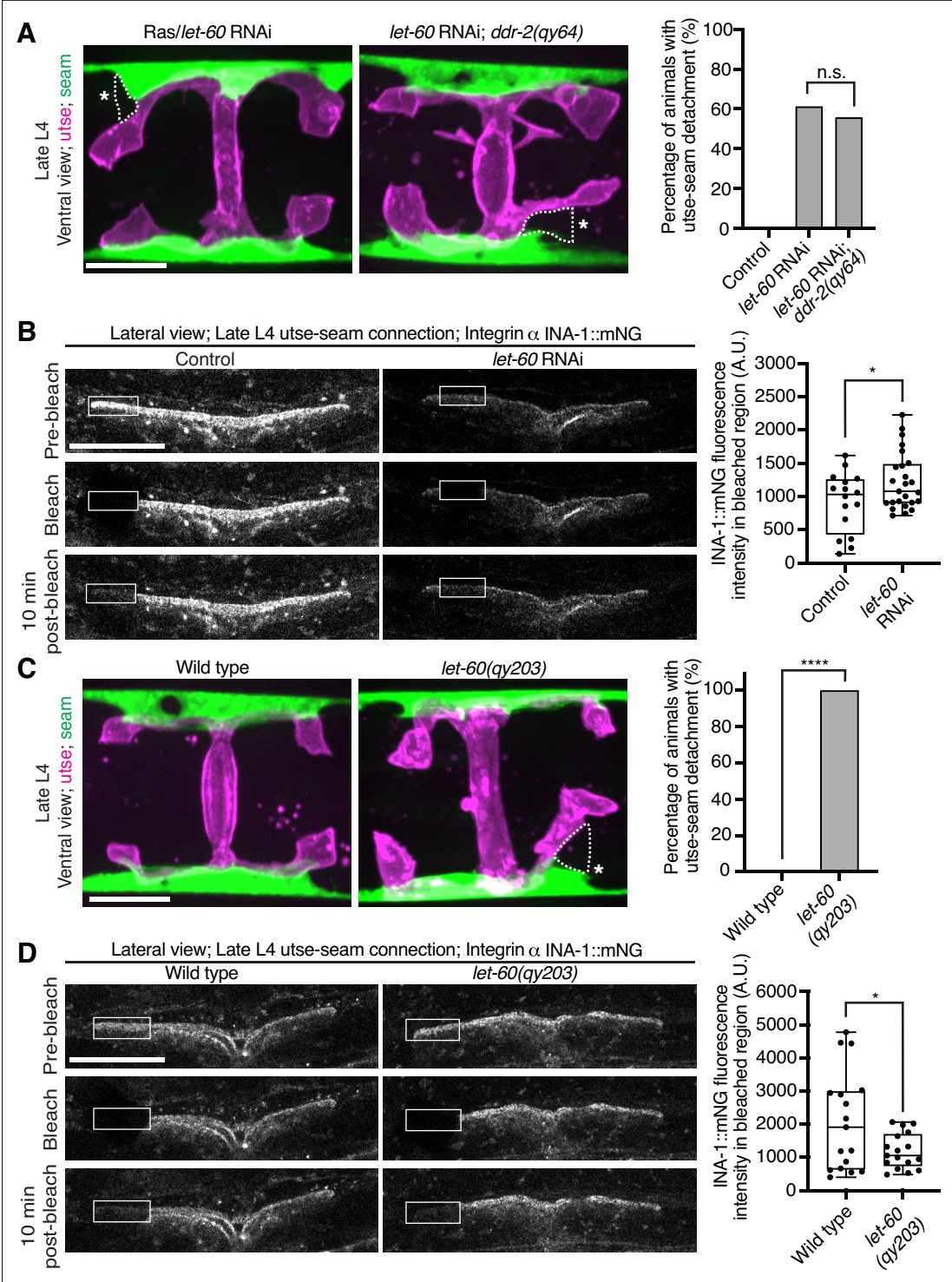

**Figure 7.** Ras/LET-60 acts in the same pathway as discoidin domain receptor-2 (DDR-2) and controls integrin stability at the utse-seam tissue connection. (**A**) Left: Ventral fluorescence z-projections of the utse (*nas-22p::2xmKate2::PLC $^{δPH}$*) and seam (*scmp::GFP::CAAX*) tissues in wild-type and *ddr-2* knockout (*ddr-2(qy64)*) animals on Ras/*let-60* RNAi treatment. Dotted lines with asterisks indicate regions of utse-seam detachment. Right: Percentage of animals with utse-seam detachment. Control n=0/21, *let-60* RNAi n=11/18, and *let-60* RNAi; *ddr-2(qy64)*, n=10/18 animals with detachments respectively. n.s. (not significant), p>0.05; Fisher's exact test. (**B**) Left: Lateral fluorescence images of INA-1::mNG at the utse-seam connection before photobleaching, immediately after photobleaching, and 10 min post-photobleaching in late L4 control or *let-60* RNAi-treated animals. Box indicates bleached region. Right: Quantification of mean INA-1::mNG fluorescence intensity in the bleached region 10 min post-photobleaching. Control n=16, *let-60* RNAi n=25. *p≤0.05, unpaired two-tailed Student's *t* test. (**C**) Left: Ventral fluorescence z-projections of the utse and seam tissues in late L4 wild-type and *let-60*

*Figure 7 continued on next page*

*Figure 7 continued*

gain-of-function mutant (*let-60(qy203)*) animals. Dotted lines with asterisks indicate regions of utse-seam detachment. Right: Quantification of utse-seam detachment frequency. Wild type, n=0/26; *let-60(qy203)*, n=11/11 animals with detachments respectively. ****p≤0.0001, Fisher's exact test. (**D**) Left: Lateral fluorescence images of INA-1::mNG at the utse-seam attachment region before photobleaching, immediately after photobleaching, and 10 min post-photobleaching in late L4 wild-type and *let-60(qy203)* animals. Box indicates bleached region. Right: Quantification of mean INA-1::mNG fluorescence intensity in the bleached region 10 min post-photobleaching (n=17 each genotype). *p≤0.05, unpaired two-tailed Student's *t* test. Scale bars, 20 µm. Box edges in boxplots represent the 25th and 75th percentiles, the line in the box denotes the median value, and whiskers mark the minimum and maximum values.

The online version of this article includes the following source data and figure supplement(s) for figure 7:

**Source data 1.** Source data for *Figure 7*.

**Figure supplement 1.** Ras/*let-60* knockdown reduces INA-1::mNG levels at the utse-seam connection.

must be strong and balanced. How cells coordinate and strengthen adhesion at BM-BM linkage sites, however, is unknown. Using the *C. elegans* utse-seam BM-BM tissue connection, we show that type IV collagen, which is the molecular glue that fastens the BM-BM linkage, also activates the *C. elegans* collagen receptor DDR-2 in the utse and seam tissues. DDR-2 activity synchronizes and bolsters an integrin-mediated adhesion to stabilize the utse-seam BM-BM tissue connection during its formation (*Figure 8*).

The linkage between the multinucleated utse cell and epithelial seam cells allows the utse to maintain the uterus within the animal during powerful egg-laying muscle contractions (*Vogel and Hedgecock, 2001*). We found that contact between the BM-encased utse and seam cells occurs at the time the utse cell is formed during the early L4 larval stage. This association happens just prior to

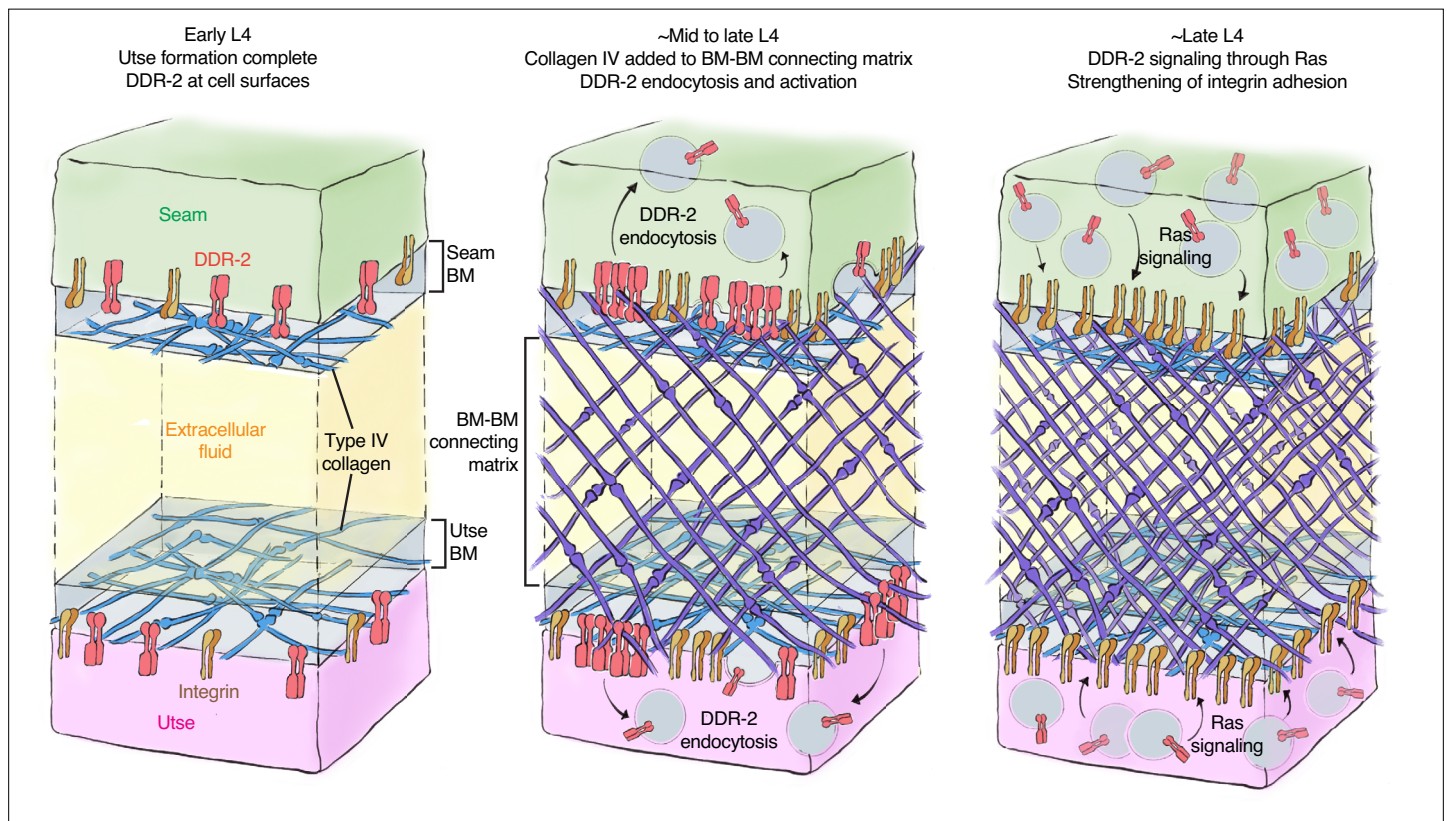

**Figure 8.** Model of discoidin domain receptor-2 (DDR-2) function at the utse-seam tissue connection. At the early L4 larval stage, DDR-2 is predominantly found at the surfaces of utse and seam cells. Between the mid and late L4 stages, type IV collagen assembles in the basement membrane (BM)-BM connecting matrix to link the utse and seam tissues. Collagen addition and binding to DDR-2 triggers DDR-2 endocytosis and activation. By the late L4, endocytic localization of DDR-2 reaches peak levels. Ras/LET-60 acts in the same pathway and may function downstream of DDR-2 to stabilize integrin adhesion. Thus, type IV collagen has a dual role in fastening the tissue connection and in signaling through DDR-2 to bolster cell adhesion at the linkage site.

assembly of type IV collagen at the BM-BM linkage, a matrix component that allows the linkage to resist the mechanical forces of egg-laying (*Gianakas et al., 2023*). Even before egg-laying, however, the utse-seam linkage resists the forces of body wall and uterine muscle contractions, as defects in matrix assembly linking the BMs results in utse-seam detachment during the mid L4 stage (*Gianakas et al., 2023*). Through analysis of genetic mutations in the *C. elegans* RTK DDR-2, an ortholog to the two vertebrate DDR receptors (DDR1 and DDR2) (*Unsoeld et al., 2013*), we discovered that loss of *ddr-2* results in utse-seam detachment beginning at the mid L4 stage. The frequency of detachments in *ddr-2* mutant animals peaked around the late L4 stage and did not increase after this time. This correlated with the levels of DDR-2::mNG at the utse-seam connection, which peaked at the late L4 stage and then sharply declined by adulthood. Together, these findings suggest that DDR-2 promotes utse-seam attachment in the early formation of the tissue connection between the mid and late L4 stage. Analysis of the BM-BM connecting matrix that links the BMs, which is composed of hemicentin, fibulin-1, and type IV collagen, revealed it was still assembled at functional levels. This suggests that the utse and seam detachment did not arise from a reduction in deposition of BM-BM linking matrix, but rather a defect in tissue adhesion to the BM-BM linkage site. Our results also indicated that DDR-2 functions in both the utse and seam to promote adhesion, as tissue-specific expression of a DN DDR-2 in either tissue resulted in utse-seam detachment. Loss of DDR1 in mice disrupts the BM-BM connection between the podocytes and endothelial cells in the kidney glomeruli and leads to defects in blood filtration (*Gross et al., 2004*), suggesting a possible shared function of DDRs in mediating proper linkage at sites of tissue connection.

In vitro studies using DDR-specific collagen binding peptides have revealed that DDRs only modestly contribute directly to cell adhesion (*Xu et al., 2012*). Instead, DDRs are thought to initiate downstream signaling cascades that promote cell adhesion (*Borza and Pozzi, 2014*). Consistent with this notion, we observed that DDR-2::mNG was not detectable at utse and seam cell surfaces when the *ddr-2* mutant attachment defect occurred between the mid to late L4 stages. Instead, DDR-2 was present in early, late, and recycling endosomes of both utse and seam cells. Work in a mouse osteoblast and a human kidney cell line has shown that DDR1 shifts from localization at the cell membrane to aggregation and internalization into early endosomes within minutes of exposure to collagen (*Mihai et al., 2009*). Examination of the timing of DDR activation has suggested that full tyrosine receptor activation occurs within the endocytic vesicles (*Mihai et al., 2009*; *Fu et al., 2013*)—a signaling hub of other RTKs (*Murphy et al., 2009*; *Villaseñor et al., 2016*). We also found that internalization of DDR-2 at the utse-seam connection correlated with the assembly of type IV collagen at the BM-BM linkage and was dependent on type IV collagen deposition. Type IV collagen is ~400 nm in length and the utse-seam connecting matrix spans ~100 nm, while the utse and seam BMs are each ~50 nm thick (*Timpl et al., 1981*; *Vogel and Hedgecock, 2001*). Thus, collagen molecules in the connecting matrix could project into the utse and seam BMs to interact with DDR-2 on cell surfaces (*Figure 8*). Consistent with this possibility, super-resolution imaging of the mouse kidney glomerular basement membrane (GBM), a tissue connection between podocytes and endothelial cells, showed type IV collagen within the GBM projecting into the podocyte and endothelial BMs (*Suleiman et al., 2013*; *Naylor et al., 2021*). As DDR-2 is activated by ligand-induced clustering of the receptor (*Juskaite et al., 2017*; *Corcoran et al., 2019*), it suggests that the BM-BM linking type IV collagen network, which is specifically assembled at high levels, clusters and activates DDR-2 in the utse and seam cells to coordinate cell-matrix adhesion at the tissue linkage site.

Vertebrate DDRs can promote integrin-mediated cell-matrix adhesion in a variety of cell types. For example, overexpression of DDR1 enhances integrin matrix adhesion in human embryonic kidney cells and in human and mouse fibroblasts (*Xu et al., 2012*; *Staudinger et al., 2013*; *Borza et al., 2022*), and high levels of DDR2 enhance integrin activation in cancer-associated fibroblasts in mice (*Bayer et al., 2019*). Several mechanisms for DDR-mediated enhancement of integrin adhesion have been proposed, including increased integrin surface levels and Rap1-mediated talin recruitment (*Staudinger et al., 2013*; *Bayer et al., 2019*). Loss of *C. elegans rap-1* has not been reported to have a Rup/uterine prolapse phenotype (*Frische et al., 2007*) and we found that DDR-2 did not promote talin recruitment to the utse-seam connection. DDR-2 did, however, increase the levels of the two *C. elegans* α integrins—INA-1 and PAT-2—at the utse-seam linkage (*Clay and Sherwood, 2015*), as well as the integrin activating protein DEB-1/vinculin (*Bays and DeMali, 2017*). Supporting a functional role for DDR-2 in promoting INA-1-mediated utse-seam attachment, *ddr-2* null mutants strongly

enhanced the utse-seam detachment and uterine prolapse defect (Rup phenotype) of animals with reduced INA-1 function. Further, like DDR-2, we found that INA-1 was expressed in the utse and seam cells and through tissue-specific expression of a DN integrin, we determined that integrin activity is required in both tissues to promote adhesion. To understand how DDR-2 affects the strength of INA-1-mediated utse-seam adhesion, we performed photobleaching experiments on endogenously tagged INA-1::mNG, and discovered that DDR-2 stabilizes INA-1-mediated adhesion. Through genetic interaction studies, RNAi, and expression of constitutively active Ras (*C. elegans* LET-60), we provide evidence that DDR-2 signals through Ras to promote INA-1 stability. Ras GTPases are a key downstream mediator of DDR signaling (*Chen et al., 2021*). Ras has been implicated in activating integrins via a poorly understood mechanism that may involve integrin receptor trafficking (*Zhang et al., 1996*; *Conklin et al., 2010*; *Sandri et al., 2012*). Interestingly, expression of the constitutively active form of Ras further increased INA-1 stability at the utse-seam linkage and led to a highly penetrant utse-seam detachment defect. Notably, our photobleaching experiments showed that there was a wide range of integrin stability at the utse-seam attachment site in wild-type animals, likely reflecting low to high adhesion strength. However, Ras activation narrowed integrin to within a high stability range (i.e., high adhesion strength), while *ddr-2* loss as well as Ras reduction shifted integrin toward a narrow, low stability range (low adhesion strength). Together, our observations suggest that DDR-2 signaling through Ras facilitates an increased range of integrin adhesion at the utse-seam connection, which is required to maintain the utse-seam linkage.

The utse-seam adhesion at the BM-BM tissue linkage is built in the L4 larval stage and then maintained in the adult to support egg-laying (*Gianakas et al., 2023*). Our evidence indicates that DDR-2 activity is highest and perhaps only required between the mid and late L4 larval stages (~ 4 hr period), as the utse-seam detachment defect peaked by the mid-to-late L4 stage and did not worsen at the late L4 and young adult stages. This timing aligns with a spike in DDR-2 levels in the utse and seam cells and is also when DDR-2 most strongly promoted INA-1 integrin and vinculin levels at the utse-seam linkage. It is possible that DDR-2 is specifically required between the mid and late L4 stages to facilitate a wide range of integrin adhesion. This could adjust the cell-matrix adhesion strength as the BM-BM connecting matrix assembles, strengthens, and changes composition during this time window, shifting from a hemicentin and fibulin-rich to a type IV collagen-dominated matrix (*Gianakas et al., 2023*). Alternatively, a range of integrin adhesion strength may be required to maintain the utse-seam linkage, while allowing the utse and seam to undergo morphogenetic changes to form the tongue (utse) and groove (seam) morphology that secures the tissue linkage. Collectively, our findings support the idea that although the ligands for DDRs are stable collagen molecules, DDRs can function as dynamic sensors of the external environment, allowing cells and tissues to coordinate their activities with changes to the extracellular surroundings.

## Materials and methods
### *C. elegans* culture and strains
*C. elegans* strains used in this study are listed in the Key resources table. All newly generated strains and associated plasmids and primers (source label 'this study' in the Key resources table) are available upon request. Worms were reared on nematode growth medium plates seeded with OP50 *Escherichia coli* at 16°C, 18°C, or 20°C according to standard procedures (*Stiernagle, 2006*). We used *C. elegans* vulval development to accurately stage animals through the morphogenesis of the utse-seam connection (late L3 to young adult stage) (*Mok et al., 2015*).

### Generation of genome-edited strains
To generate the genome-edited mNeonGreen (mNG) knock-in allele for *deb-1* (*qy48*), we used CRISPR/Cas9 genome editing with a self-excising cassette (SEC) for drug selection (hygromycin treatment) as described previously (*Dickinson et al., 2013*; *Keeley et al., 2020*). An 18 amino acid flexible linker attached to the mNG fluorophore was inserted in frame and directly upstream of the stop codon. sgRNA sequences directing Cas9 cleavage near the C-terminus are provided in the Key resources table.

The *ddr-2* knockout allele *ddr-2(qy64)*, an 8615 bp deletion corresponding to the coding sequence of DDR-2, was also generated with the SEC CRISPR method detailed above with one modification—we

generated a new starter repair plasmid lacking the fluorescent tag. Primers used to amplify homology arms, sgRNA sequences to direct Cas9 cleavage, and genotyping primers used to verify genome-edited knockout animals are provided in the Key resources table.

For the LET-60[G13E] gain-of function mutation, *let-60(qy203)*, we used the same strategy as the *ddr-2* knockout allele. The point mutation was incorporated into the primer for homology arm amplification. Note that this mutation is identical to the *let-60(n1046)* allele.

## Generation of transgenic strains

For the utse marker *qy91 [nas-22p::2xmKate2::pH]*, we generated a single-copy transgene inserted into the ttTi5605 transposon insertion site on chromosome II (*Frøkjær-Jensen et al., 2012*) using the SEC CRISPR method detailed above. A 2 kb *nas-22* promoter, two tandem mKate2 fluorophores attached to a PH domain, and an *unc-54* 3' UTR fragment were ligated in order via Gibson assembly into the pAP087 starter repair plasmid containing homology arms for the ttTi5605 site. The sgRNA directing Cas9 cleavage near this region is contained in the pDD122 vector.

To express mKate2-tagged RAB-5 in the seam, we generated a single-copy transgene inserted into the ttTi4348 transposon insertion site on chromosome I (*Frøkjær-Jensen et al., 2012*) using SEC CRISPR. Briefly, the scm promoter, mKate2 fluorophore, rab-5 genomic sequence, and *unc-54* 3' UTR were Gibson assembled in order into the pAP088 starter repair plasmid containing homology arms for the ttTi4348 site. The sgRNA directing Cas9 cleavage near this region is contained in the pCFJ352 vector.

For tissue-specific expression of DN DDR-2 in the utse or seam, we also generated single-copy transgenes inserted into the ttTi4348 site. For utse expression, the following fragments were Gibson assembled in order into pAP088: a 1.5 kb *cdh-3* promoter fragment, *DN ddr-2*, mNG fluorophore, and *unc-54* 3' UTR. Expression in transgenic animals was verified by assessing mNG fluorescence. For seam expression, the following fragments were Gibson assembled in order into pAP088: the *scm* promoter fragment, *DN ddr-2*, and mKate2 fluorophore. Expression in transgenic animals was verified by assessing mKate2 fluorescence.

The seam cell marker *qyEx605 [scmp::2xmKate2::PH]* was built by Gibson assembling a 1.3 kb *scm* promoter fragment, a fragment with two tandem mKate2 fluorophores attached to the PH domain, and *unc-54* 3' UTR fragment in order. The construct was expressed as extrachromosomal arrays as described previously (*Yochem and Herman, 2003*).

To build the construct for *DN integrin* expressed in the seam, we first amplified the *DN integrin* fragment from animals harboring the *qyIs15* allele (see Key resources table). The *DN integrin* fragment is composed of a *pes-10* enhancer element, the *ost-1* signal peptide, the *pat-3* β-tail fragment, and *unc-54* 3' UTR as described previously (*Lee et al., 2001*). The *DN integrin* fragment was fused to a 1.3 kb *wrt-2* promoter by PCR. The construct was expressed extrachromosomally.

## RNAi

All RNAi constructs except for *ddr-2* RNAi were obtained from the Ahringer and Vidal RNAi libraries (*Kamath and Ahringer, 2003*; *Rual et al., 2004*) or generated previously (*Gianakas et al., 2023*). For the *ddr-2* clone, an ~2.4 kb cDNA fragment corresponding to the longest *ddr-2* transcript was amplified by PCR and inserted into the T444t vector as described previously (*Gianakas et al., 2023*). RNAi experiments were performed using the feeding method (*Timmons et al., 2001*) according to previously detailed protocols (*Jayadev et al., 2019*; *Gianakas et al., 2023*). For experiments assessing the utse-seam connection, RNAi was initiated in synchronized L1 larvae and animals were examined at the late L4 stage (~44 hr treatment). For examination of uterine prolapse, RNAi was performed from the L1 through adulthood (~120 hr treatment). For depletion of hemicentin initiated at the early L4 stage, synchronized L1 worms were grown on control RNAi for ~38 hr to the early L4, and then transferred onto *him-4* RNAi to the late L4 (~5–6 hr treatment). We verified knockdown efficiencies for all RNAi experiments except for *let-60* RNAi by including a control with the relevant fluorescent tagged target protein, achieving between ~75% and 100% reduction. We verified *let-60* RNAi knockdown efficiency by plate level assessment of brood size, which was strongly reduced compared to control animals, as previously reported (*Ceron et al., 2007*).

## Microscopy and image processing

All fluorescence images (except for photobleaching experiments) were acquired at 20°C on Zeiss Axio Imager A1 microscopes controlled by the µManager software v.1.4.23 or v.2.0.1 (*Edelstein et al., 2010*), equipped with a Hamamatsu ImagEM electron multiplying charge-coupled device camera or an Orca-Fusion scientific complementary metal oxide semiconductor camera, Zeiss 40× and 100× Plan Apochromat (1.4 numerical aperture) oil immersion objectives, Yokogawa CSU-10 spinning disc confocal scan heads, and 488 nm, 505 nm, and 561 nm laser lines. Worms were mounted on 5% noble agar pads containing 0.01 M sodium azide for imaging.

For photobleaching experiments, images were acquired at 20°C on a Zeiss 880 single-point scanning confocal attached to a Zeiss Axio Observer Z1 microscope, with a Marzhauser linearly encoded stage, a 40× Plan Neofluar (1.3 numerical aperture) oil immersion objective, and a 488 nm laser line. Worms were anesthetized by soaking in 5 mM levamisole in M9 buffer for 15 min and then transferred to 4% noble agar pads. Coverslips were then placed and sealed on top with valap (equal weight Vaseline, lanolin, and paraffin) and flooded with 5 mM levamisole. Coverslip sealing kept the animals hydrated during the experimental timeframe, ensuring accurate imaging of endogenous protein dynamics (detailed in *Kelley et al., 2017*). A rectangle region within one arm of the BM-BM connection BLINK was photobleached using 30 iterations of simultaneous 405 nm and 488 nm excitation at 100% laser power for a total bleaching time of 1.5 s. Worms were imaged prior and immediately following bleaching, and then again 10 min later.

Brightfield images of worms were acquired at 20°C on a Zeiss Axio Zoom V16 stereo fluorescence microscope controlled by Zen 3.2 software, equipped with an Axiocam digital camera and a 3× objective. Worms were imaged without immobilization.

To image the utse and seam cells ventrally, we manually oriented animals in the ventral orientation (*Kelley et al., 2017*) and acquired z-stacks at 0.37 µm intervals, capturing the entire H-shaped structure of the utse and the seam cells flanking both sides of the utse at ×100 magnification. Ventral views of the utse and seam shown in figures are maximum intensity projections of these z-stacks. 3D isosurface renderings of the utse and seam cells shown in *Figure 2D* were generated with Imaris 7.4 software (Bitplane). We acquired lateral z-stacks at 0.37 µm intervals to fully capture the superficial half of the utse-seam attachment.

For lateral imaging of DDR-2::mNG together with the utse, seam cells, or endosome markers expressed in the utse or seam, we acquired z-stacks at 0.37 µm intervals to capture as much of each tissue that was in focus superficially at ×40 magnification. Images shown in figures are single slices where DDR-2 was sharply in focus within the tissue of interest. The same parameters were used for imaging INA-1::mNG and PAT-2::mNG in the utse and seam. Images in *Figure 5A* correspond to the z-slices where INA-1 and PAT-2 were most in focus at the BM-BM connection region. In *Figure 5— figure supplement 1A and B*, images shown are sum projections of the seam or utse regions.

To image DDR-2::mNG specifically at the utse-seam attachment site, we acquired lateral z-stacks at 0.37 µm intervals capturing both the surface of the utse and the seam cells. We used a utse cell marker as a reference to accurately capture this region (see also schematic in *Figure 3A*). Images shown in *Figure 4* are sum projections of these z-stacks.

For lateral imaging of respective tagged proteins at the BM-BM connection region, we acquired single slices (×40 magnification) at the middle focal plane where the BM-BM connection signal was sharply in focus. All images were processed in Fiji 2.0 (*Schindelin et al., 2012*).

## Image analysis and quantification

To assess utse-seam detachment, we used maximum intensity projections of the utse and seam viewed ventrally as detailed above. We defined utse-seam detachment as any gap in the utse-seam attachment of at least 4 square pixels in area. We preferred the ventral view over 3D rendering for this metric as gaps were variable in size and location (typically occurring in one or two arms of the utse on opposite sides). The ventral projections allowed us to reliably examine all four arms of the utse and whether they were in contact with the seam from animal to animal. The visual representation of a gap in a ventral maximum projection is an underestimate of the true size of the detachment, as the tongue-in-groove association of the utse and seam can only be seen in a 3D surface rendering of the utse and seam viewed from the side. We thus used a binary scoring system (animals with or without utse-seam detachments) instead of quantifying the size of the detached region. We note that ventral imaging

of the utse and seam sometimes results in small utse-seam detachments as we manually orient the animals on the slide. To control for this, we acquired large datasets (at least 40 animals examined) for every control strain we used to visualize the utse-seam attachment with. We found that small detachments related to ventral imaging occurred in ~8–12% of animals, depending on the genetic background. All datasets shown in figures have been normalized to account for this.

All quantifications of mean fluorescence intensity were done on raw images in Fiji 2.0. We drew ~5-pixel long and 2-pixel wide linescans to obtain raw values of mean fluorescence intensity. We measured mean fluorescence intensity for all quantifications in order to account for linescan area. For measurement within the BM-BM connection region, we positioned the linescan within the arm that was most in focus. Other regions where linescans were performed are indicated in figures. Background intensity values were obtained with similar linescans in adjacent regions with no visible fluorescence signal. Note that we used two tandem mNG fluorophores to visualize PAT-2 at the utse-seam attachment site (PAT-2::2xmNG) as it was present at very low levels. We imaged PAT-2::mNG and PAT-2::2xmNG in this region at the same exposure and found that the latter was approximately two times brighter (PAT-2::2xmNG mean fluorescence intensity 3941±847 AU, PAT-2::mNG 2037±465 AU, n=10 animals examined each). Taken together with quantifications in *Figure 5B*, INA-1 is thus likely present at sixfold higher levels than PAT-2 at the utse-seam attachment site. For additional quantification related to FRAP, see below.

For FRAP analysis, we measured mean fluorescence intensity within the bleached region. In addition, we performed an equivalent linescan within the unbleached arm at the BM-BM connection to calculate a bleach correction factor to account for general photobleaching during image acquisition across the duration of the experiment, as previously described (*Gianakas et al., 2023*). Briefly, the background-corrected fluorescence intensity measurement on the side of the BM-BM connection where FRAP was not performed at the 10 min post-bleach timepoint was divided by the respective value at the pre-bleach timepoint to obtain the bleach correction factor. Fluorescence intensity measurements at the pre-bleach and post-bleach timepoints on the side of the BM-BM connection where FRAP was performed were then multiplied by the respective bleach correction factor to normalize these values to the 10 min post-bleach fluorescence intensity measurement in the same region.

To count the number of DDR-2 punctae at the site of utse-seam attachment, we used sum projections of DDR-2::mNG capturing the surface of the utse and seam (described earlier). We restricted measurements to the utse-seam connection corresponding to one utse arm. Brightness and contrast was then adjusted to manually count individual punctae, as DDR-2 punctae showed a large variance in size and fluorescence intensity. Note that in the type IV collagen depletion condition, we only counted discrete puncta and did not include the large cell surface accumulation of DDR-2 in the analysis.

To determine co-localization of DDR-2 with markers of endocytic vesicles, we also used a manual approach due to the large variability of the size and fluorescence intensity of punctae. We selected a single slice where the utse or the seam was sharply in focus, and restricted measurements to a single utse arm or a region of the seam corresponding to a single utse arm. For each animal, the number and positions of DDR-2 punctae were noted and compared to the respective endosome marker (RAB-5, RAB-7, or RAB-11). If a DDR-2 puncta physically overlapped with or was in contact with an endosome punctum, it was classified as one that co-localized with the endosomal marker. We then calculated the percentage overlap of DDR-2 punctae with the vesicle marker for each animal (corresponding to each data point within boxplots in *Figure 3—figure supplement 1B and C*).

## Scoring of uterine prolapse

Uterine prolapse frequency was assessed as described previously (*Gianakas et al., 2023*). Briefly, synchronized L1 larvae were plated (~20 animals per plate) and after 24 hr, the exact number of worms on each plate was recorded. Plates were then visually screened for ruptured worms (uterine prolapse) every 24 hr during egg-laying (between 48 hr and 120 hr post-L1). We chose to examine the entire egg-laying period as ruptures arising from utse-seam detachments do not usually occur at the onset of egg-laying, but after cycles of egg-laying that place repeated mechanical stress on the utse-seam connection (*Gianakas et al., 2023*). Ruptured animals were scored and removed from plates to avoid double counting and after 120 hr, the percentage of animals with uterine prolapse was calculated. At least 50 animals were screened for every experiment.

### Illustrations of utse-seam tissue connection

For schematics in *Figure 1A* and *Figure 8*, model protein sizing was scaled according to previously generated models (*Keeley et al., 2020*). We sized hemicentin, fibulin, type IV collagen, integrin, and DDR-2. We also approximated the span of the BMs of the utse and seam cells (~50 nm each), as well as the BM-BM connecting matrix (~200 nm), based on electron microscopy of the utse-seam attachment (*Vogel and Hedgecock, 2001*).

### Statistical analysis

Statistical analysis was performed in GraphPad Prism 9. At least two independent biological replicates were performed for every experiment. Sample sizes were validated a posteriori by comparing the spread of data between individual trials. All data shown in graphs were pooled from respective individual trials whose distributions did not differ significantly. Normality of datasets was assessed using the D'Agostino-Pearson normality test. We used parametric tests for datasets that followed a Gaussian distribution and non-parametric tests for those that did not. For comparisons of means between two populations, we used either an unpaired two-tailed Student's $t$ test or a Mann-Whitney $U$ test. For comparisons of means between three or more populations, we performed either a one-way ANOVA followed by post hoc Dunnett's test or Kruskal-Wallis $H$ test with post hoc Dunn's test. For comparisons between two categorical variables, we used the Fisher's exact test. All graphs were prepared in GraphPad Prism. Figure legends indicate sample sizes, statistical tests used, and p values.

## Acknowledgements

We would like to thank C Borza, A Pozzi, B Hoffman, and D Reiner for helpful discussions, C Gianakas for comments on the manuscript, and M Boxem and H Hutter for strains. Some strains were provided by the CGC, which is funded by NIH Office of Research Infrastructure Programs (P40 OD010440). SGP was supported by graduate research fellowship NICHD F31 HD97901. This work was supported by R35GM118049, R21OD028766, and R21OD032430 to DRS.

## Additional information

### Funding

| Funder | Grant reference number | Author |
|---|---|---|
| National Institute of General Medical Sciences | R35GM118049 | David R Sherwood |
| National Institutes of Health | R21OD028766 | David R Sherwood |
| National Institutes of Health | R21OD032430 | David R Sherwood |
| National Institutes of Health | F31 HD97901 | Sara G Payne |

The funders had no role in study design, data collection and interpretation, or the decision to submit the work for publication.

### Author contributions

Kieop Park, Ranjay Jayadev, Conceptualization, Data curation, Formal analysis, Validation, Investigation, Visualization, Methodology, Writing - original draft, Writing - review and editing; Sara G Payne, Conceptualization, Data curation, Formal analysis, Funding acquisition, Validation, Investigation, Visualization, Methodology, Writing - original draft, Writing - review and editing; Isabel W Kenny-Ganzert, Qiuyi Chi, William Ramos-Lewis, Investigation, Methodology; Daniel S Costa, Methodology; Siddharthan B Thendral, Visualization; David R Sherwood, Conceptualization, Supervision, Funding acquisition, Writing - original draft, Project administration, Writing - review and editing

## Author ORCIDs

Ranjay Jayadev (iD) http://orcid.org/0000-0003-0465-0337
David R Sherwood (iD) http://orcid.org/0000-0002-4448-6917

Reviewer #1 (Public Review): https://doi.org/10.7554/eLife.87037.3.sa1
Reviewer #2 (Public Review): https://doi.org/10.7554/eLife.87037.3.sa2
Author Response: https://doi.org/10.7554/eLife.87037.3.sa3

---

# Additional files

## Supplementary files

• MDAR checklist

## Data availability

All data generated in this study are included in the manuscript and supporting files.

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

# Appendix 1

## Appendix 1—key resources table

| Reagent type (species) or resource | Designation | Source or reference | Identifiers | Additional information |
|---|---|---|---|---|
| Strain, strain background (*C. elegans*) | NK2617 | *Gianakas et al., 2023* | | qyIs23 [cdh-3p::mCh::PH] II; lqIs80 [scmp::GFP::CAAX] IV |
| Strain, strain background (*C. elegans*) | N2 | Caenorhabditis Genetics Center (CGC) | | Wild-type (ancestral) |
| Strain, strain background (*C. elegans*) | RB970 | Caenorhabditis Genetics Center (CGC) | | ddr-1(ok874) X |
| Strain, strain background (*C. elegans*) | VH1387 | *Unsoeld et al., 2013* | | ddr-1(tm382) X |
| Strain, strain background (*C. elegans*) | RB788 | Caenorhabditis Genetics Center (CGC) | | ddr-2(ok574) X |
| Strain, strain background (*C. elegans*) | VH1383 | *Unsoeld et al., 2013* | | ddr-2(tm797) X |
| Strain, strain background (*C. elegans*) | NK2511 | This study | | ddr-2(qy64) X |
| Strain, strain background (*C. elegans*) | NK2655 | This study | | qyIs23 [cdh-3p::mCh::PH] II; lqIs80 [scmp::GFP::CAAX] IV; ddr-2(qy64) X |
| Strain, strain background (*C. elegans*) | NK2640 | This study | | qy91 [nas-22p::2xmKate2::PH] II; lqIs80 [scmp::GFP::CAAX] IV |
| Strain, strain background (*C. elegans*) | NK2682 | This study | | qy91 [nas-22p::2xmKate2::PH] II; lqIs80 [scmp::GFP::CAAX] IV; ddr-2(qy64) X |
| Strain, strain background (*C. elegans*) | NK2994 | This study | | qy44 [ddr-2::mNG] X; qyEx605 [scmp::2xmKate2:PH] |
| Strain, strain background (*C. elegans*) | NK2620 | This study | | qyIs23 [cdh-3p::mCh::PH] II; qy44 [ddr-2::mNG] X |
| Strain, strain background (*C. elegans*) | NK2763 | This study | | qyIs256 [cdh-3p::mCherry::rab-5]?; qy44 [ddr-2::mNG] X |
| Strain, strain background (*C. elegans*) | NK2766 | This study | | qyIs252 [cdh-3p::mCherry::rab-7]?; qy44 [ddr-2::mNG] X |
| Strain, strain background (*C. elegans*) | NK2769 | This study | | qyIs205 [cdh-3p::mCherry::rab-11]?; qy44 [ddr-2::mNG] X |
| Strain, strain background (*C. elegans*) | NK2901 | This study | | qy188 [scmp::mKate2::rab-5] I; qy44 [ddr-2::mNG] X |
| Strain, strain background (*C. elegans*) | NK2926 | This study | | qy91 [nas-22p::2xmKate2::PH] II; lqIs80 [scmp::GFP::CAAX] IV; ddr-2(ok574) X |
| Strain, strain background (*C. elegans*) | NK2705 | *Jayadev et al., 2022* | | qy118 [col-99::mNG (internal tag)] IV |
| Strain, strain background (*C. elegans*) | NK2322 | *Keeley et al., 2020* | | qy22 [cle-1::mNG] I |
| Strain, strain background (*C. elegans*) | NK2326 | *Keeley et al., 2020* | | qy24 [emb-9::mNG (internal tag)] III |
| Strain, strain background (*C. elegans*) | NK2830 | This study | | qy166 [cdh-3p::dominant negative ddr-2::mNG] I; qy91 [nas-22p::2xmKate2::PH] II; heIs63 [wrt-2p::GFP::PH +wrt-2p::GFP::H2B+lin-48p:mCherry] V |
| Strain, strain background (*C. elegans*) | NK2848 | This study | | qy175 [scmp::dominant negative ddr-2::mKate2] I; qy91 [nas-22p::2xmKate2::PH] II; heIs63 [wrt-2p::GFP::PH +wrt-2p::GFP::H2B+lin-48p:mCherry] V |
| Strain, strain background (*C. elegans*) | NK2585 | *Jayadev et al., 2022* | | qy83 [emb-9::mRuby2 (internal tag)] III |

*Appendix 1 Continued on next page*

*Appendix 1 Continued*

| Reagent type (species) or resource | Designation | Source or reference | Identifiers | Additional information |
|---|---|---|---|---|
| Strain, strain background (*C. elegans*) | NK2422 | *Keeley et al., 2020* | | qy33 [him-4::mNG] X |
| Strain, strain background (*C. elegans*) | NK2324 | *Jayadev et al., 2019* | | qy23 [ina-1::mNG] III |
| Strain, strain background (*C. elegans*) | NK2479 | *Jayadev et al., 2019* | | qy49 [pat-2::2xmNG] III |
| Strain, strain background (*C. elegans*) | NK2825 | This study | | qy23 [ina-1::mNG] III; ddr-2(qy64) X |
| Strain, strain background (*C. elegans*) | NK2858 | This study | | qy49 [pat-2::2xmNG] III; ddr-2(qy64) X |
| Strain, strain background (*C. elegans*) | NK2804 | This study | | qyIs23 [cdh-3p::mCh::PH] II; heIs63 [wrt-2p::GFP::PH +wrt-2p::GFP::H2B+lin-48p:mCherry] V |
| Strain, strain background (*C. elegans*) | NK2824 | This study | | qyIs23 [cdh-3p::mCh::PH] II; qyIs15 [zmp-1$^{mk50-51}$p::dominant negative integrin b-pat-3] IV;heIs63 [wrt-2p::GFP::PH +wrt-2p::GFP::H2B+lin-48p:mCherry] V |
| Strain, strain background (*C. elegans*) | NK2934 | This study | | qy91 [nas-22p::2xmKate2::PH] II; unc-119(ed4) III; Iqls80 [scmp::GFP::CAAX] IV; qyEx604 [wrt-2p::dominant negative integrin b-pat-3+unc-119(+)] |
| Strain, strain background (*C. elegans*) | NK2579 | *Keeley et al., 2020* | | qy62 [mNG::fbl-1] IV |
| Strain, strain background (*C. elegans*) | NK932 | *Keeley and Sherwood, 2019* | | qyIs190 [zmp-4p::zmp-4::GFP] |
| Strain, strain background (*C. elegans*) | NK268 | *Hagedorn et al., 2009* | | qyIs17 [zmp-1$^{mk50-51}$p mCherry] |
| Strain, strain background (*C. elegans*) | AH3437 | *Walser et al., 2017* | | zh117 [GFP::tln-1] I |
| Strain, strain background (*C. elegans*) | NK2478 | This study | | qy48 [deb-1::mNG] IV |
| Strain, strain background (*C. elegans*) | NK2854 | This study | | zh117 [GFP::tln-1] I; ddr-2(qy64) X |
| Strain, strain background (*C. elegans*) | NK2860 | This study | | qy48 [deb-1::mNG] IV; ddr-2(qy64) X |
| Strain, strain background (*C. elegans*) | NK2944 | This study | | qy91 [nas-22p::2xmKate2::PH] II; Iqls80 [scmp::GFP::CAAX] IV; let-60(qy203) IV |
| Strain, strain background (*C. elegans*) | NK2957 | This study | | qy23 [ina-1::mNG] III; let-60(qy203) IV |
| Strain, strain background (*E. coli*) | emb-9 RNAi | *Kamath and Ahringer, 2003* | | Clone from Ahringer library (L4440 vector backbone) |
| Strain, strain background (*E. coli*) | col-99 RNAi | *Rual et al., 2004* | | Clone from Vidal library (L4440 vector backbone) |
| Strain, strain background (*E. coli*) | cle-1 RNAi | *Rual et al., 2004* | | Clone from Vidal library (L4440 vector backbone) |
| Strain, strain background (*E. coli*) | ina-1 RNAi | *Rual et al., 2004* | | Clone from Vidal library (L4440 vector backbone) |
| Strain, strain background (*E. coli*) | let-60 RNAi | *Rual et al., 2004* | | Clone from Vidal library (L4440 vector backbone) |
| Strain, strain background (*E. coli*) | deb-1 RNAi | *Rual et al., 2004* | | Clone from Vidal library (L4440 vector backbone) |
| Strain, strain background (*E. coli*) | him-4 RNAi | *Gianakas et al., 2023* | | T444t vector backbone |

*Appendix 1 Continued on next page*

*Appendix 1 Continued*

| Reagent type (species) or resource | Designation | Source or reference | Identifiers | Additional information |
|---|---|---|---|---|
| Strain, strain background (*E. coli*) | ddr-2 RNAi | This study | | T444t vector backbone |
| Sequence-based reagent | ddr-2 knockout (ddr-2(qy64)) sgRNA 1 | This study | | ATCCTGACATAGATGAGCGT |
| Sequence-based reagent | ddr-2 knockout (ddr-2(qy64)) sgRNA 2 | This study | | GTCATTGGTGCACACTTCTC |
| Sequence-based reagent | ddr-2 knockout (ddr-2(qy64)) sgRNA 3 | This study | | AAGTGTGCACCAATGACTGG |
| Sequence-based reagent | deb-1::mNG (deb-1(qy48)) sgRNA 1 | This study | | AGTTGGACCACATTGGCTTT |
| Sequence-based reagent | deb-1::mNG (deb-1(qy48)) sgRNA 2 | This study | | ATTTAGAAGTTGGACCACAT |
| Sequence-based reagent | let-60 gain-of-function (let-60(qy203)) sgRNA 1 | This study | | CTTGTGGTAGTTGGAGATGG |
| Sequence-based reagent | Primer ddr-2(qy64) homology arm forward | This study | | TTTTCAGAGTCTCCGACGCTCATCTA |
| Sequence-based reagent | Primer ddr-2(qy64) homology arm reverse | This study | | TAAATATTATTCTGAGAATATA |
| Sequence-based reagent | Primer ddr-2(qy64) genotyping forward | This study | | TGGTAATTGATGAGAGGGTG |
| Sequence-based reagent | Primer ddr-2(qy64) genotyping reverse | This study | | TGTCGTTTCGACACCGGCAA 1.8 kb band |
| Sequence-based reagent | Primer let-60(qy203) homology arm forward | This study | | ATGACGGAGTACAAGCTTGTG GTAGTTGGAGATGGAGAAGT |
| Sequence-based reagent | Primer let-60(qy203) homology arm reverse | This study | | TACCCTTTTCTGAAAAAAGACGC |
| Sequence-based reagent | Primer nas-22 promoter forward | This study | | TCAAAGCGTCAAGCTTTACG |
| Sequence-based reagent | Primer nas-22 promoter reverse | This study | | TTTTCTGTCTACAAGGACTGTGC |
| Sequence-based reagent | Primer scm promoter forward | *Chapman et al., 2008* | | AACCTCCACCAGATGGTTGGCG |
| Sequence-based reagent | Primer scm promoter reverse | *Chapman et al., 2008* | | CCCGGGGATCCGTCCACTCT |
| Sequence-based reagent | Primer cdh-3 promoter forward | This study | | CTAGAGCATGATGTCCTTAC |
| Sequence-based reagent | Primer cdh-3 promoter reverse | This study | | CAAAACGGACCGACCGTCCC |
| Sequence-based reagent | Primer dominant negative ddr-2 forward | This study | | ATGAAGTTGCTGCTGTATCT |
| Sequence-based reagent | Primer dominant negative ddr-2 reverse | This study | | TCTGCTCACGCAAATCAACT |
| Sequence-based reagent | Primer wrt-2 promoter forward | This study | | TCAGAACTCTAATACTTACT |
| Sequence-based reagent | Primer wrt-2 promoter reverse | This study | | CCGAGAAACAATTGGCAGGT |
| Sequence-based reagent | Primer dominant negative integrin b pat-3 forward | *Hagedorn et al., 2009* | | TCTAGAGGATCCCGGGGAT |
| Sequence-based reagent | Primer dominant negative integrin b pat-3 reverse | *Hagedorn et al., 2009* | | ATTTAGTTGGCTTTTCCAGC |

*Appendix 1 Continued on next page*

*Appendix 1 Continued*

| Reagent type (species) or resource | Designation | Source or reference | Identifiers | Additional information |
|---|---|---|---|---|
| Sequence-based reagent | Primer ddr-2 RNAi forward | This study | | ATGAAGTTGCTGCTGTATCT |
| Sequence-based reagent | Primer ddr-2 RNAi reverse | This study | | ATGAATATGAGGAGAAGTGTGC |
| Recombinant DNA reagent | Plasmid: pCFJ352 | Addgene | RRID: Addgene_30539 | sgRNA targeting MosSCI insertion site on Chr I, from Erik Jorgensen |
| Recombinant DNA reagent | Plasmid: pDD122 | *Dickinson et al., 2013* | RRID: Addgene_47550 | sgRNA targeting MosSCI insertion site on Chr II |
| Recombinant DNA reagent | Plasmid: pAP087 | *Pani and Goldstein, 2018* | | Starter SEC repair template plasmid for single copy insertion at the ttTi5605 site on chromosome II |
| Recombinant DNA reagent | Plasmid: pAP088 | *Pani and Goldstein, 2018* | | Starter SEC repair template plasmid for single copy insertion at the ttTi4348 site on chromosome I |
| Chemical compound, drug | Ampicillin | Sigma-Aldrich | #A0166 | |
| Chemical compound, drug | Isopropyl β-D-1-thiogalactopyranoside | Sigma-Aldrich | #I6758 | |
| Chemical compound, drug | Hygromycin B | Sigma-Aldrich | #H3274 | |
| Chemical compound, drug | Sodium azide | Sigma-Aldrich | #S2002 | |
| Chemical compound, drug | Levamisole | Sigma-Aldrich | #L9756 | |
| Software, algorithm | μManager v.1.4.23 | *Edelstein et al., 2010* | RRID: SCR_016865 | |
| Software, algorithm | Zen Black | Zeiss | RRID:SCR_018163 | |
| Software, algorithm | Fiji/ImageJ | *Schindelin et al., 2012* | RRID:SCR_002285 | |
| Software, algorithm | GraphPad Prism v9 | GraphPad Software | RRID:SCR_002798 | |

