## [Editor Report · eLife assessment]

This **important** paper reveals how cells in adjacent tissues use the extracellular matrix to establish mechanical connections. Through a series of crisp genetic manipulations and quantitative image analyses, the authors provide **compelling** evidence to show how an essential adhesion between the uterus and the seam cells in the nematode *C. elegans* is formed. The assembly of type IV collagen triggers internalization of a cell surface receptor, which then signals from endocytic vesicles to strengthen the connection.

---

## [Referee Report · Reviewer #1 (Public Review)]

Park et al demonstrate that cells on either side of a BM-BM linkage strengthen their adhesion to that matrix using a positive feedback mechanism involving a discoidin domain receptor (DDR-2) and integrin (INA-1 + PAT-3). In response to its extracellular ligand (Collagen IV/EMB-9), DDR-2 is endocytosed and initiates signaling that in turn stabilizes integrin at the membrane. DDR-2 signaling operates via Ras/LET-60. This work's strength lies in its excellent in vivo imaging, especially of endogenously tagged proteins. For example, tagged DDR-2:mNG could be seen relocating from seam cell membranes to endosomes. I also think a second strength of this system is the ability to chart the development of BM-BM linkage over time based on the stages of worm larval development. This allows the authors to show DDR signaling is needed to establish linkage, rather than maintain it. It likely is relevant to many types of cells that use integrin to adhere to BM and left me pondering a number of interesting questions. For example: (1) Does DDR-2 activation require integrin? Perhaps integrin gets the process started and DDR-2 positively reinforces that (conversely is DDR-2 at the top of a linear pathway)? (2) In ddr-2(qy64) mutants, projections seem to form from the central portion of the utse cell. Does this reveal a second function for DDR-2, regulating perhaps the cytoskeleton? And (3) can you use the forward genetic tools available in *C. elegans* to find new genes connecting DDR-2 and integrin? The authors discuss these ideas in their response to the reviews, and I look forward to hearing about their future work on these questions.

I do see two areas where the manuscript could be improved. First, the authors rely on imprecise genetic methods to reach their conclusions (i.e. systemic RNAi, or expression of dominant negative constructs.) I think their conclusion would be stronger if they used tissue specific degradation to block ddr-2 function specifically in the utse or seam cells. Methods to do this are now regularly used in *C. elegans* and the authors have already developed the necessary tissue-specific promoters. Second, the manuscript is presented in the introduction as a study on formation and function of BM-BM linkage. However, their results actually demonstrate a mechanism by which cells adhere to BM. Since ddr-2 appears to function equally in both utse + seam cells (based on their dominant negative data), there are likely three layers of adhesion (utse-BM, BM-BM, BM-seam) and if any of those break down, you get a partially penetrant rupture phenotype. I pointed this out in my initial review, and after reading the revised manuscript, I do still feel the authors' introduction presents the paper as dealing with how basement membranes link together. But, I wonder if this might this be a question of terminology/language use? Maybe I am operating on a strict definition of linkage, and the authors use it more inclusively. What term(s) should we use to differentiate two basement membranes that are linked together, versus tissues that are connected through a basement membrane linkage? This is something that could be clarified in future publications.

These concerns do not undercut the significance of this work, which identifies an interesting mechanism cells use to strengthen adhesion during BM linkage formation. In fact, I am excited to read future papers detailing the connection between DDR-2 and integrin. But before undertaking those experiments the authors should be certain which cells require DDR-2 activity, and that should not be determined based solely on mis expression of a dominant negative.

---

## [Referee Report · Reviewer #2 (Public Review)]

This paper explores the mechanisms by which cells in tissues use the extracellular matrix (ECM) to reinforce and establish connections. This is a mechanistic and quantitative paper that uses imaging and genetics to establish that the Type IV collagen, DDR-2/collagen receptor discoidin domain receptor 2, signaling through Ras to strengthen an adhesion between two cell types in *C. elegans*. This connection needs to be strong and robust to withstand the pressure of the numerous eggs that pass through the uterus. The major strengths of this paper are in crisply designed and clear genetic experiments, beautiful imaging, and well supported conclusions. I find very few weaknesses, although, perhaps the evidence that DDR-2 promotes utse-seam linkage through regulation of MMPs could be stronger. This work is impactful because it shows how cells in vivo make and strengthen a connection between tissues through ECM interactions involving collaboration between discoidin and integrin.

---

## [Author Response]

The following is the authors' response to the original reviews.

We have now incorporated the changes recommended by the reviewers to improve the interpretations and clarity of the manuscript. We are grateful for their thoughtful comments and suggestions, which have significantly strengthened the manuscript.

**Reviewer #1 (Public Review):**
Park et al demonstrate that cells on either side of a BM-BM linkage strengthen their adhesion to that matrix using a positive feedback mechanism involving a discoidin domain receptor (DDR-2) and integrin (INA-1 + PAT-3). In response to its extracellular ligand (Collagen IV/EMB-9), DDR-2 is endocytosed and initiates signaling that in turn stabilizes integrin at the membrane. DDR-2 signaling operates via Ras/LET-60. This work's strength lies in its excellent in vivo imaging, especially of endogenously tagged proteins. For example, tagged DDR-2:mNG could be seen relocating from seam cell membranes to endosomes. I also think a second strength of this system is the ability to chart the development of BM-BM linkage over time based on the stages of worm larval development. This allows the authors to show DDR signaling is needed to establish linkage, rather than maintain it. It likely is relevant to many types of cells that use integrin to adhere to BM and left me pondering a number of interesting questions.

We thank the reviewer for highlighting the strengths and impact of our work in expanding our understanding of tissue linkages and how DDR and integrins might work in other contexts.

For example: (1) Does DDR-2 activation require integrin? Perhaps integrin gets the process started and DDR-2 positively reinforces that (conversely is DDR-2 at the top of a linear pathway)?

DDR activation by receptor clustering upon exposure to its ligand collagen is well documented (Juskaite et al., 2017 *eLife* PMID: 285ti0245). Clustered DDR is rapidly internalized into endocytic vesicles, where full activation of tyrosine kinase activity is thought to occur (Fu et al., 2013 *J Biol Chem* PMID: 23335507). Supporting this model, we found that concentrated type IV collagen is required for vesicular DDR-2 localization in the utse and seam cells at the utse-seam connection. Whether DDR-2 activation requires integrin has not been fully established. However, one study using mouse and human cell lines showed that DDR1 activation occurs independent of integrin (Vogel et al., 2000 *J Biol Chem* PMID: 10681566), consistent with the latter possibility raised by the reviewer that DDR-2 is upstream of integrin.

To test these hypotheses, we require an experimental condition where loss or near complete loss of INA- 1 integrin is achieved by the mid-to-late L4 larval stage, when DDR-2 is activated by collagen and taken into endocytic vesicles. Currently, we can only partially deplete INA-1 by RNAi (Figure 5—ﬁgure supplement 2E), and strong loss of function mutations in *ina-1* result in early larval arrest and lethality (Baum and Garriga, 1titi7 *Neuron* PMID: ti247263). To overcome these obstacles, we are adapting the new FLP-ON::TIR1 system developed for precise spatiotemporal protein degradation in worms (Xiao et al., 2023 *Genetics* PMID: 36722258). We hope to achieve a near complete knockdown of *ina-1* with this timed depletion strategy. In the future, we will use this system to block DDR-2 and integrin function speciﬁcally in the utse or seam cells, to complement our current dominant negative mis-expression approach.

(2) In ddr-2(qy64) mutants, projections seem to form from the central portion of the utse cell. Does this reveal a second function for DDR-2, regulating perhaps the cytoskeleton?

We thank the reviewer for their observation and agree with their interpretation. We think it is important to comment on this and have stated in the results text, lines 208-212: “In addition, membrane projections emanating from the central body of the utse were detected in *ddr-2(qy64)* animals. These projections were ﬁrst observed at the mid L4 stage and persisted to young adulthood (Figure 2C). These observations suggest that DDR-2 functions around the mid L4 to late L4 stages to promote utse-seam attachment, and that DDR-2 may also regulate utse morphology.”

And (3) can you use the forward genetic tools available in *C. elegans* to find new genes connecting DDR-2 and integrin?

This is an excellent suggestion. We found that loss of *ddr-2* strongly enhanced the uterine prolapse (Rup) defect caused by RNAi mediated depletion of integrin. To ﬁnd new genes connecting DDR-2 and integrin, a targeted screen for the Rup phenotype could be performed in an integrin reduction of function condition. As we cannot work with null or strong loss-of-function *ina-1* alleles (described above), the screen could be conducted with either timed depletion of INA-1 with candidate RNAi treatments, or combinatorial *ina-1* RNAi with candidate RNAi treatments.

I do see two areas where the manuscript could be improved. First, the authors rely on imprecise genetic methods to reach their conclusions (i.e. systemic RNAi, or expression of dominant negative constructs.) I think their conclusion would be stronger if they used tissue specific degradation to block ddr-2 function specifically in the utse or seam cells. Methods to do this are now regularly used in *C. elegans* and the authors have already developed the necessary tissue-specific promoters.

We agree with the reviewer that tissue speciﬁc degradation of DDR-2 in the utse and seam cells will complement and strengthen our evidence for the site of action of DDR-2. As described earlier, we are currently adapting the FLP-ON::TIR1 tissue degradation system to perform these experiments and will provide our ﬁndings in a follow-up manuscript.

Second, the manuscript is presented in the introduction as a study on formation and function of BM-BM linkage. The authors start the discussion in a similar manner. But their results are about adhesion between cells and BM. In fact they show the BM-BM linkage forms normally in ddr-2 mutants. Thus it seems like what they have really uncovered is an adhesion mechanism that works in parallel to the BM-BM linkage. Since ddr-2 appears to function equally in both utse + seam cells (based on their dominant negative data), there are likely three layers of adhesion (utse-BM, BM-BM, BM-seam) and if any of those break down, you get a partially penetrant rupture phenotype.

The reviewer raises an important and interesting point, and we agree that we did not articulate the organization of the utse-seam tissue connection clearly. The utse-seam connection is comprised of the utse and seam BMs each ~50nm thick, and a connecting matrix bridging the two BMs, which is ~100nm thick (Vogel and Hedgecock, 2001 *Development* PMID: 11222143). Type IV collagen builds up to high levels within the connecting matrix and links the utse and seam BMs, and its concentration is required for DDR-2 vesiculation. An important point we did not highlight is that type IV collagen is approximately 400 nm long (Timpl et al. 1ti81, *Eur J Biochem* PMID: 6274634). Thus, collagen molecules within the connecting matrix could span the entire length of the utse-seam connection and project into the utse and seam BMs to interact with cell surface receptors. Consistent with this possibility, we found that buildup of type IV collagen that spans the utse-seam BM-BM linkage correlated with the timing of DDR-2 activation/vesiculation within utse and seam cells. In addition, super-resolution imaging of the mouse kidney glomerular basement membrane (GBM), a tissue connection between endothelial BM and epithelial (podocyte) BM, showed type IV collagen, which spans the BMs, projects into the endothelial and podocyte BMs (Suleiman et al., 2013 eLife PMID: 24137544). We carefully considered these points to generate the schematics in Figure 1A and Figure 8, but failed to articulate this point in the manuscript. We are grateful for the reviewer for bringing up our error and have now stated these details in the text to address the reviewer’s concern as outlined below.

In the introduction (lines ti3-ti6): “A BM-BM tissue connection between the large, multinucleated uterine utse cell and epidermal seam cells stabilizes the uterus during egg laying. The utse-seam connection is formed by BMs of the utse and the seam cells, each ~50 nm thick, which are bridged by an ~100 nm connecting matrix (Vogel and Hedgecock 2001, Morrissey, Keeley et al. 2014, Gianakas, Keeley et al. 2023).”

In the discussion (lines 507-520): “We also found that internalization of DDR-2 at the utse-seam connection correlated with the assembly of type IV collagen at the BM-BM linkage and was dependent on type IV collagen deposition. Type IV collagen is ~400 nm in length and the utse-seam connecting matrix spans ~100 nm, while the utse and seam BMs are each ~50 nm thick (Timpl, Wiedemann et al. 1ti81, Vogel and Hedgecock 2001). Thus, collagen molecules in the connecting matrix could project into the utse and seam BMs to interact with DDR-2 on cell surfaces. Consistent with this possibility, super- resolution imaging of the mouse kidney glomerular basement membrane (tiBM), a tissue connection between podocytes and endothelial cells, showed type IV collagen within the tiBM projecting into the podocyte and endothelial BMs (Suleiman, Zhang et al. 2013). As DDR-2 is activated by ligand-induced clustering of the receptor (Juskaite, Corcoran et al. 2017, Corcoran, Juskaite et al. 201ti), it suggests that the BM-BM linking type IV collagen network, which is speciﬁcally assembled at high levels, clusters and activates DDR-2 in the utse and seam cells to coordinate cell-matrix adhesion at the tissue linkage site.”

These concerns do not undercut the significance of this work, which identifies an interesting mechanism cells use to strengthen adhesion during BM linkage formation. In fact, I am excited to read future papers detailing the connection between DDR-2 and integrin. But before undertaking those experiments the authors should be certain which cells require DDR-2 activity, and that should not be determined based solely on mis expression of a dominant negative.

We thank the reviewer for recognizing the signiﬁcance of our work and reiterate that we will use tissue-speciﬁc degradation for site of action experiments in future studies on the biology of the utse- seam tissue linkage.

**Reviewer #2 (Public Review):**
This paper explores the mechanisms by which cells in tissues use the extracellular matrix (ECM) to reinforce and establish connections. This is a mechanistic and quantitative paper that uses imaging and genetics to establish that the Type IV collagen, DDR-2/collagen receptor discoidin domain receptor 2, signaling through Ras to strengthen an adhesion between two cell types in *C. elegans*. This connection needs to be strong and robust to withstand the pressure of the numerous eggs that pass through the uterus. The major strengths of this paper are in crisply designed and clear genetic experiments, beautiful imaging, and well supported conclusions. I find very few weaknesses, although, perhaps the evidence that DDR-2 promotes utse-seam linkage through regulation of MMPs could be stronger. This work is impactful because it shows how cells in vivo make and strengthen a connection between tissues through ECM interactions involving collaboration between discoidin and integrin.

We appreciate the reviewer’s assessment of the impact of our work in detailing a mechanism for how cells increase their adhesion to the ECM to establish connections between adjacent tissues. We have softened the interpretation of our MMP localization data to address the reviewer’s concern (detailed below).

**Reviewer #1 (Recommendations For The Authors):**
Regarding Figure 1D, is it possible to show when the BM forms on the cartoons more clearly (something like the 3rd section of Fig 3A)? I can see it in the timeline but it's hard to follow in the diagrams.

We agree with the reviewer that we could show when the BM-BM connecting matrix forms more clearly in Figure 1D. Hemicentin and ﬁbulin, the earliest components of the connecting matrix, are detected at very low levels at the utse-seam connection during the mid-L4 stage and are more prominently localized by the mid-to-late L4 stage (Gianakas et al., 2023 *J Cell Biol* PMID: 36282214). For this reason, we only show the connecting matrix in yellow from the mid-to-late L4 stages onward. We have now made the BM-BM connection more prominent in the ﬁgure 1D cartoons with boxed outlines (similar to Figure 3A as the reviewer suggested). We also added a label for the time window when the BM-BM connection forms.

Regarding the RNAi induced prolapse phenotype, looking at 2B, it appears that between 5% and 10% of animals have uterine prolapse when fed control RNAi. Is this correct, it seems very high? This prolapse in control animals was not observed other RNAi experiments such as Figure 5C.

We thank the reviewer for pointing this out. For Figure 2B, the control used was wild-type N2 animals fed with OP50 *E. coli* bacteria, rather than HT115 bacteria carrying the L4440 empty vector (control RNAi). This is because the main comparisons were to ﬁve *ddr-1* and *ddr-2* mutant strains. We did notice a slightly higher baseline uterine prolapse frequency (5% on average, detailed in Figure 2—Source data 1) in wild-type animals fed OP50 bacteria, compared to HT115 bacteria fed animals (approximately 1-2% on average). It is possible this could be linked to the nutritional diﬀerences in the two bacterial strains. However, we are conﬁdent of our data in Figure 2B as we carried out 3 independent trials, and the uterine prolapse frequencies in *ddr-1* mutant animals matched the baseline in wild-type animals, while the frequencies for *ddr-2* mutants were all increased over the baseline in all trials (as detailed in Figure 2—Source data 1).

Relating to the point above, in reading the methods to try to understand how they did the RNAi, I noticed that they measure prolapse continually over five days. I didn't realize it takes a long time to occur. I think they should explain this in the text and in the figures. Reading the manuscript I thought prolapse occurred as soon as mutant animals began laying eggs. In the text they should explain this when they first assay the phenotype (page 7), and for figures the Y axis on the graphs could say "% uterine prolapse after 5 days."

We thank the reviewer for their suggestions. We did not articulate clearly that the utse-seam connection is able to withstand some mechanical stress, even when key components are lost. It’s only over time and repeated use that the connection breaks down. This is likely because a number of components contribute to the connection and as we have shown previously, there is feedback, such that when one components is reduced, such as collagen, hemicentin is increased in levels at the BM-BM connection. Since ruptures arising from utse-seam detachments typically occur sometime after the onset

of egg-laying, we screened the entire egg-laying period (days two to ﬁve post-L1) as described in Gianakas et al. 2023. We have now incorporated these points in the text and ﬁgures as follows:

In the introduction, we clariﬁed that utse-seam BM-BM connection breaksdown over time, by adding (lines titi-105): “Hemicentin promotes the recruitment of type IV collagen, which accumulates at high levels at the BM-BM tissue connection and strengthens the adhesion, allowing it to resist the strong mechanical forces of egg-laying. The utse-seam connection is robust, with each component of the tissue- spanning matrix contributing to the BM-BM connection (Gianakas, Keeley et al. 2023). This likely accounts for the ability of the utse-seam connection to initially resist mechanical forces after loss of any one of these components, delaying the uterine prolapse phenotype until sometime after the initiation of egg-laying.”

We expanded the results text when we ﬁrst describe the Rup phenotype (lines 183-184): “We ﬁrst screened for the Rup phenotype caused by uterine prolapse, observing animals every day during the egg-laying period, from its onset (48 h post-L1) to end (120 h) (Methods)”.

We provided more detail in the Methods section (lines 784-7ti0): “Uterine prolapse frequency was assessed as described previously (Gianakas et al 2023). Brieﬂy, synchronized L1 larvae were plated (~20 animals per plate) and after 24 h, the exact number of worms on each plate was recorded. Plates were then visually screened for ruptured worms (uterine prolapse) every 24 h during egg-laying (between 48 h to 120 h post-L1). We chose to examine the entire egg-laying period as ruptures arising from utse-seam detachments do not usually occur at the onset of egg-laying, but after cycles of egg-laying that place repeated mechanical stress on the utse-seam connection (Gianakas et al 2023).”

Finally, we modiﬁed the Y-axes of graphs in Figure 2B and 5C and the respective ﬁgure legends as suggested by the reviewer.

Then I went back and compared to the previous publication (Gianakas, 2023). I would be interested to see a time course of how many animals prolapse after 1 day, 2 days, etc.? Is this consistent with their data on hemicentrin?

We agree with the reviewer that a time course of uterine prolapse would be interesting as we saw ruptures occur throughout the egg-laying period. However, for the hemicentin knockdown experiments in Gianakas et al. 2023 as well as the experiments in this study, we recorded only the pooled number of animals with ruptures at the end of the experimental window. In future studies we will also record the uterine prolapse frequencies on each day to generate time courses that will provide more insight into the function of proteins at the utse-seam connection.

Lines 183-184: I'm not sure what it means to say "trended towards displaying a significant Rup phenotype?" Since the difference was not statistically significant, it would be better to say something like "increased but not statistically significant."

We agree with the reviewer and have now modiﬁed this sentence (lines 190-193): “Animals carrying the *ddr-2(ok574)* allele, which deletes a portion of the intracellular kinase domain (Unsoeld, Park et al. 2013),also showed an increased frequency of the Rup phenotype compared to wild-type animals, although this diﬀerence was not statistically signiﬁcant (Figure 2A and B)”.

Line 186: 'penetrant' needs a qualifier to indicate the magnitude of the proportion of individuals with the phenotype.

As we provide the Rup frequency numbers in Figure 2—Source data 1, we modiﬁed the sentence as follows (lines 1ti3-1ti5): “We further generated a full-length *ddr-2* deletion allele, *ddr-2(qy64)*, and conﬁrmed that complete loss of *ddr-2* led to a signiﬁcant uterine prolapse defect (Figure 2A and B).”

Lines 206-208; could the mounting/imaging procedure (which I assume requires squeezing the worm between agarose pad and coverslip) alter the occurrence of prolapse? I would think prolapse would occur more frequently under these conditions as compared to worms laying eggs on a plate.

The reviewer brings up an important concern. The mounting and imaging procedure does require placing the worm between an agarose pad and a coverslip. However, this did not alter the occurrence of uterine prolapse in this experiment. We were careful to perform the same procedure on both wild-type and *ddr- 2(qy64)* animals to control for this. As detailed in the manuscript, none of the eight wild-type animals we mounted underwent uterine prolapse after recovery oﬀ the coverslip, and among the *ddr-2(qy64)* mutants we mounted, only the ones that exhibited utse-seam detachments went on to rupture later.

We articulated these points more clearly by modifying lines 214-216 as follows: “Wild-type and *ddr- 2(qy64)* animals were mounted and imaged at the L4 larval stage for utse-seam attachment defects, recovered, and tracked to the 72-hour adult stage, where they were examined for the Rup phenotype.”

In seam cells you can see that DDR-2:mNG is present at membranes from early to mid L4, which makes sense. But I cannot see it on the membrane at any time point in the utse. Perhaps it is obscured by the yellow dotted line. Should it be visible on utse membranes before it is endocytosed?

The reviewer raises an interesting question. We think it is likely that DDR-2 is initially on the membrane of the utse like it is on the seam cells. However, we have not observed this, possibly due to the complex shape and thin membrane extensions of the utse. We are unable even to detect clear membrane enrichment of membrane markers in the utse (for example, compare the utse and seam membrane markers in Figure 3B). Thus, we refrained from speculating on DDR-2 utse membrane localization in the manuscript, and instead focused on the pattern of vesicular DDR-2 peaking at the late L4 stage, which was clearly visible in both the utse and seam cells.

Sup Fig 3A - please show quantification of seam cells not contacting utse at the same Y-axis scale as for regions that do contact utse.

We have modiﬁed the Y-axis scale for the quantiﬁcation of the seam region not contacting the utse.

Figure 4A - I don't see a difference between WT and ok574 - what am I missing?

In the representative *ok574* animal shown, a portion of the utse arm on the top right is detached from the seam. To make this phenotype clearer, we have recropped the image panels, readjusted the brightness and contrast of the utse and the seam, and redrawn the outline of the detachment to make this clearer.

Figure 4C+D, and lines 296-298: I'd bet that both are needed to recruit DDR-2 to membranes. But him-4 has a more severe phenotype because the RNAi knockdown is much more effective (perhaps b/c they are using the newer t444t vector).

We agree with the reviewer that the *him-4* knockdown phenotype is likely more severe than *emb-9* knockdown. Type IV collagen at the utse-seam connection is very stable compared to hemicentin (Gianakas et al 2023, *J Cell Biol* PMID: 36282214, see Fig. 5C), which could explain the lower knockdown eﬃciency.

We modiﬁed our interpretation of the data in the text as follows (lines 308-312): “In addition, we did not detect DDR-2 at the cell surface, suggesting that hemicentin has a role in recruiting DDR-2 to the site of utse-seam attachment. It is possible that collagen could also function in DDR-2 recruitment, but we could not assess this deﬁnitively due to the lower knockdown eﬃciency of *emb-9* RNAi (Figure 4—ﬁgure supplement 1A).”

**Reviewer #2 (Recommendations For The Authors):**
Line 218 DDR-2 (typo)

We have corrected this typo.

Evidence (line 344-348) may not be strong enough to say whether or not DDR-2 promotes utse- seam linkage through regulation of MMPs.

We agree with the reviewer and have softened our conclusions as follows (lines 356-363): “The *C. elegans* genome harbors six MMP genes, named zinc metalloproteinase 1-6 (zmp-1-6) (Altincicek, Fischer et al. 2010). We examined four available reporters of ZMP localization (ZMP-1::tiFP, ZMP-2::tiFP, ZMP-3::tiFP, and ZMP-4::tiFP) (Kelley, Chi et al. 201ti).Only ZMP-4 was detected at the utse-seam connection and its localization was not altered by knockdown of *ddr-2* (Figure 5—ﬁgure supplement 1F). These observations suggest that DDR-2 does not promote utse-seam linkage through regulation of MMPs, although we cannot rule out roles for DDR-2 in promoting the expression or localization of ZMP-5 or ZMP-6.”

The authors show the critical period is in late L4, however, is the signaling needed later too? For example, is the linkage strengthening moderated by DDR-2 important as more eggs accumulate?

The reviewer raises an interesting question. We observed that the vesicular localization of DDR-2 sharply declined before the onset of egg-laying. By young adulthood, very few punctate structures of DDR-2 were observed in the seam cells, and none in the utse (Figure 3B). Furthermore, the frequency of utse- seam detachments in *ddr-2* mutant animals peaked by the late L4 stage and did not increase after this time, suggesting DDR function is no longer required after the late L4 stage (Figure 2D). Thus, we believe that DDR-2 signaling strengthens tissue linkage only during the early formation of the utse-seam connection between the mid and late L4 stage.

We incorporated these points in the discussion (lines 477-485): “Through analysis of genetic mutations in the *C. elegans* receptor tyrosine kinase (RTK) DDR-2, an ortholog to the two vertebrate DDR receptors (DDR1 & DDR2) (Unsoeld, Park et al. 2013), we discovered that loss of *ddr-2* results in utse-seam detachment beginning at the mid L4 stage. The frequency of detachments in *ddr-2* mutant animals peaked around the late L4 stage and did not increase after this time. This correlated with the levels of DDR-2::mNG at the utse-seam connection, which peaked at the late L4 stage and then sharply declined by adulthood. Together, these ﬁndings suggest that DDR-2 promotes utse-seam attachment in the early formation of the tissue connection between the mid and late L4 stage.”

Fig. 3B is the ﬂuorescence quantiﬁcation normalized to the area?

Yes, it is. We used mean ﬂuorescence intensity for all ﬂuorescence quantiﬁcations to normalize for the area where the signal was measured. We added a line in Methods to emphasize this (lines 73ti-740): “We measured mean ﬂuorescence intensity for all quantiﬁcations in order to account for linescan area.”

Fig. 4B a statistical assessment of the degree of co-localization of DDR-2::mNG and the endosomal markers might be a nice addition.

We believe the reviewer is referring to Figure 3—ﬁgure supplement 1B. We have now added the statistical assessment of the degree of co-localization of DDR-2::mNG and the endosomal markers.

We want to sincerely thank the two reviewers for their thoughtful comments and suggestions. The changes we have made in response to these comments have substantially improved the manuscript.